# A double-box model for aircraft exhaust plumes based on the MADE3 aerosol microphysics (MADE3 v4.0)

Monica Sharma<sup>1,2</sup>, Mattia Righi<sup>1</sup>, Johannes Hendricks<sup>1</sup>, Anja Schmidt<sup>1,3</sup>, Daniel Sauer<sup>1</sup>, and Volker Grewe<sup>1,2</sup>

**Correspondence:** Monica Sharma (monica.sharma@dlr.de)

Abstract. Aviation emissions of aerosol particles and aerosol precursor gases alter the Earth's radiation budget via both direct and indirect aerosol effects, resulting in a significant climate effect. Current estimates of aviation-induced climate effects are based on coarse-resolution global aerosol-climate models, which are not able to resolve the microphysical processes at the aircraft plume scale. This results in large uncertainties on the aviation-induced impact on aerosol number and size, which are key quantities for estimating the aerosol indirect effect, especially for low-level liquid-phase clouds. In this work, a double-box aircraft exhaust plume model is developed to explicitly simulate the aerosol microphysics inside a dispersing aircraft exhaust plume, together with a simplified representation of the vortex regime (which begins  $\sim 10$  s after the aircraft emissions and captures the dynamics of aerosol particle interactions with contrail ice particles). The aircraft exhaust plume model is used to quantify the aviation-induced aerosol number concentration at the end of the dispersion regime ( $\sim 46$  h) and the results are compared with the result obtained by the instantaneous dispersion approach commonly applied by the global models. The difference between the plume approach (simulated using two boxes) and the instantaneous dispersion approach (simulated by a single box) is defined as the plume correction: for typical cruise conditions over the North Atlantic and typical aviation emission parameters, the plume correction for aviation-induced particle number concentration ranges between -15% and -4.2%, depending on the presence or absence of the contrail ice in the vortex regime, respectively. A tendency-based process analysis shows that the negative value of the plume correction is due to the higher efficiency of coagulation and nucleation processes in the plume approach, leading to lower total particle number concentrations compared to the instantaneous dispersion approach. Sensitivity studies over different regions highlight the role of background conditions for the plume microphysics, with the plume correction varying between -12% for Europe and -42% for China in a scenario with contrail ice in the vortex regime. Parametric studies performed on various aviation emission parameters used to initialise the plume model demonstrate the high relevance of contrail ice in the vortex regime to considerably reduce the aviation-induced aerosol number concentration in the plume approach. Moreover, the parametric studies show a large sensitivity towards aviation fuel sulfur content, driving sulfur dioxide (SO<sub>2</sub>) emissions and the sulfuric acid (H<sub>2</sub>SO<sub>4</sub>) formation, which in turn is a primary driver for the nucleation process. Thanks to its flexible configuration and minor additional computational costs, the plume model presented here can readily be applied in coarse-resolution global aerosol-climate models or used as offline parametrisation to quantify the climate effects of aviation-induced aerosol particles.

<sup>&</sup>lt;sup>1</sup>Deutsches Zentrum für Luft- und Raumfahrt (DLR), Institut für Physik der Atmosphäre, Oberpfaffenhofen, Germany

<sup>&</sup>lt;sup>2</sup>Delft University of Technology, Aerospace Engineering, Operations & Environment, 2629 HS Delft, The Netherlands

<sup>&</sup>lt;sup>3</sup>Meteorological Institute, Ludwig Maximilian University of Munich, Munich, Germany

## 1 Introduction

Aviation is contributing to climate change by emitting carbon dioxide (CO<sub>2</sub>), nitrogen oxides (NO<sub>x</sub>=NO+NO<sub>2</sub>), water vapour, aerosol particles and their precursors. The aviation-induced effective radiative forcing in 2018 as an indicator for its climate effects since 1940 is estimated to be 3.5% of the total anthropogenic forcing (Lee et al., 2021). With an increasing demand of commercial air transportation, the fuel consumption is expected to increase, which will result in a rise in global aviation emissions in the future (Esmeijer et al., 2020; Grewe et al., 2021). About one third of the aviation-induced climate effect is attributable to CO<sub>2</sub> emissions, while the non-CO<sub>2</sub> emissions are considered responsible for the remaining two thirds (Lee et al., 2021, 2023). However, the uncertainties associated with this estimate are large, in particular when it comes to contrail radiative forcing and aerosol indirect effects. The aircraft exhaust emitted at typical cruise altitude (9-13 km) consists of a mixture of CO<sub>2</sub> and non-CO<sub>2</sub> compounds (Penner et al., 1999). The latter are released in the form of gases (e.g., NO<sub>x</sub> and SO<sub>2</sub>) and aerosols, mainly sulfate and soot. Amongst the non-CO<sub>2</sub> emissions, aerosol particles are known to significantly affect Earth's radiation budget by scattering and absorbing incoming solar radiation and by perturbing the microphysical and radiative properties of clouds. Recent studies argued that aviation-induced aerosol particle emissions may be potentially relevant not only for high-level cirrus clouds (Hendricks et al., 2011; Penner et al., 2018; Righi et al., 2021), but also for low-level clouds in the liquid phase (Gettelman and Chen, 2013; Righi et al., 2013; Kapadia et al., 2016; Righi et al., 2023). However, the related microphysical processes and the resulting climate effects are highly uncertain. One of the reasons for these uncertainties is the representation of the aerosol microphysical processes, primarily coagulation, condensation and nucleation, which control the aerosol properties at the plume scale, in particular their number concentration and size. Such plume-scale processes cannot be resolved by the global aerosol-climate models due to their coarse spatial resolution, limiting their ability to simulate changes in aviation-induced particle number concentrations, which directly controls the formation of cloud droplets at lower altitudes and hence the aviation-induced effects on cloud properties and lifetime.

At the typical cruise altitude for a commercial aircraft the evolution of the aircraft exhaust plume is categorized in three main regimes based on their time of emission and other chemical and microphysical processes, namely the jet regime, the vortex regime and the dispersion regime (Kärcher, 1995; Fritz et al., 2020; Unterstrasser et al., 2014; Tait et al., 2022). The first and shortest regime of an aircraft plume is jet regime and it lasts about 10 seconds. In this regime the hot and humid exhaust, with extremely high concentrations of aviation-induced species being present at the engine exit at temperatures of around 700 K to 1000 K mixes with the ambient air. During the jet regime contrail ice can be formed under specific atmospheric conditions (Schmidt, 1941; Appleman, 1953; Schumann, 1996). Subsequently, in the vortex regime, the emitted exhaust is trapped inside the wake vortices formed behind the aircraft. The vortex regime can last for a few minutes. During this phase, the plume cools while attaining thermal equilibrium and reaches the typical cruise ambient temperature (~220 K). The dispersion regime is the longest regime of aircraft exhaust plume and it can last from several hours to days (Paoli et al., 2011; Fritz et al., 2020). In the dispersion regime, the aerosol particles undergo several chemical and microphysical processes, such as particle coagulation, as well as sedimentation. If the emitted particles survive in sufficient numbers by the end of the dispersion regime, they can be transported towards lower atmospheric layers where they may act as cloud condensation nuclei (CCNs) in liquid

clouds, hence contributing to the aerosol indirect effects (Gettelman and Chen, 2013; Righi et al., 2013). During the dispersion regime, the plume expands both laterally and vertically behind the aircraft following turbulent mixing with the background air (Schumann et al., 1995). Existing global aerosol-climate models simply assume the aircraft exhaust to mix homogeneously and instantaneously with the background air in a large-scale grid-cell, neglecting essential processes at the plume scale (Brasseur et al., 1998; Cariolle et al., 2009; Paoli et al., 2011; Fritz et al., 2020; Tait et al., 2022). This is an intrinsic limitation of global aerosol-climate models due to their coarse spatial resolution (~100 km), making it unfeasible to follow the sub-grid scale non-linear plume processes of emitted aerosol particles in the expanding and dispersing aircraft exhaust plume (Meilinger et al., 2005), which in turn affects the simulated aerosol number concentration of aviation-induced particles, resulting in large uncertainties in their climate effect on low-level clouds (Lee et al., 2021). The stated limitations may explain the large range in current-generation global model-based estimates of the effective radiative forcing (ERF) from the interactions of aviation-aerosol with low-level clouds, ranging between –164 mW m<sup>-2</sup> (Gettelman and Chen, 2013) to –22 mW m<sup>-2</sup> (Kapadia et al., 2016). The given values are also sensitive to other aircraft parameters, such as for instance the sulfur content of the jet fuel (Kapadia et al., 2016), and to the assumed size of the emitted aerosol particles (Gettelman and Chen, 2013; Righi et al., 2013).

To overcome the limitations in representing microphysical processes at the plume scale in coarse-resolution aerosol-climate models, we developed a double-box aircraft exhaust plume model that explicitly accounts for the aerosol microphysical processes in the expanding and dispersing aircraft plume. Specifically, the plume model allows to quantify the impact of these processes on the resulting aerosol particle properties, such as mass, number and size, at the end of the plume dispersion. The particle properties are then compared with those obtained by the instantaneous dispersion approach adopted by coarse resolution global models. The relative difference between the plume approach and the instantaneous dispersion approach is defined as the plume correction. This correction can later be applied in global models in order to account for the unresolved plume processes in such models. The plume model is based on the aerosol microphysics of the well-established MADE3 aerosol submodel (Modal Aerosol Dynamics model for Europe, adapted for global applications, 3rd generation; Kaiser et al., 2014), which is extended into a two-box configuration to represent the aircraft plume and the surrounding background, respectively, as well as their interaction. This approach allows to explicitly simulate the aerosol microphysical processes within the dispersing aircraft plume, while accounting for the diffusion dynamics and the entrainment of background air into the dispersing plume. The plume model also features a simple, physically based representation of the aerosol coagulation with contrail ice crystals during the vortex regime.

Several existing models are capable of simulating microphysics and gas-phase chemistry of fine aerosol particles (Binkowski and Shankar, 1995), including the chemistry of particles in the aircraft plumes and their transformation via dispersion and microphysical processes (Kärcher et al., 1996; Brown et al., 1996; Petry et al., 1998; Kraabøl and Stordal, 2000; Meilinger et al., 2005; Fritz et al., 2020). The plume model developed here specifically addresses the limitations of global models to simulate the impact of aviation aerosol, by comparing the instantaneous approach typical of those models with a more sophisticated representation of the aerosol processes at the plume scale. We do not aim to develop a plume dynamic model to simulate the eddies in the wake vortices as done, for instance, by Unterstrasser et al. (2014). The main goal of this work is to develop a

plume model that can improve the representation of aerosol microphysical processes inside a dispersing aircraft plume, which are relevant for the indirect effect of aviation emitted aerosols and the resulting climate effects.

The plume model concept and structure is described in Sect. 2. In Sect. 3 we present a first application for typical cruise conditions over the North Atlantic and the model sensitivity to key processes, like coagulation with contrail ice and particle nucleation. We also explore the impact of different background conditions in different regions and the sensitivity of the results to different assumptions for the initial parameters. We finally discuss the limitations of our approach, the perspectives for applying the plume model as a parametrization in global model studies and the possible implications for climate assessments (Sect. 4). The key conclusions of this work are summarized in Sect. 5.

# 2 Double-box aircraft exhaust plume model description

## 2.1 Concept






The concept of the double-box aircraft exhaust plume model developed in this study is based on Petry et al. (1998), who originally introduced it to better account for the plume chemistry of the reactive species inside an aircraft exhaust plume at cruise altitude. The plume model introduced in our study explicitly focuses on simulating the aerosol microphysics and on comparing the results of instantaneous mixing of the aircraft exhaust inside a large-scale grid-box with a more detailed approach where the diffusion of a dispersing plume within the background is simulated in detail. By comparing the resulting aerosol mass and number concentrations at the end of the plume dispersion, we assess the effects of utilizing the more precise plume dispersion approach compared to an instantaneous dispersion approach. In the present work, the concept of Petry et al. (1998) is further adapted to account for the aerosol microphysical processes within a dispersing aircraft plume and extended to further include the interactions of aerosol with the ice crystals during the vortex regime of the plume evolution.

In the instantaneous dispersion approach usually adopted by global models, the aircraft emissions are instantaneously distributed over the large (~100 km) grid-box and homogeneously mixed at once. This method completely disregards the non-linear microphysical processes occurring at plume scale, thus misrepresenting the impact of key transformation processes (such as particle coagulation and nucleation) taking place during the expansion and dispersion of the plume in the surrounding background (Gustafson et al., 2011). This may in turn lead to inaccurate estimates of the aviation-induced particle properties at the end of the dispersion regime, in particular in terms of aerosol number concentration and size (Gettelman and Chen, 2013; Righi et al., 2013; Tait et al., 2022). These properties are critical in the context of the climate effect of aviation aerosol, as they control the potential perturbation on cloud droplet number concentration in liquid clouds and hence the resulting radiative forcing via aerosol indirect effects. The approach developed here allows to explicitly simulate the plume scale processes representing the aerosol microphysics inside a dispersing aircraft plume. Fig. 1 schematically describes our approach: the plume model is applied to simulate the dispersion of a single aircraft plume (represented by the single-plume box, hereafter SP) inside a background atmosphere (represented by the background box, BG). This plume approach is then compared with the instantaneous dispersion approach, where the aviation emissions are instantaneously released in the background of a single box (ID box). The microphysics in all three boxes (SP, BG and ID) is simulated using the aerosol sub-model MADE3, in its box model

configuration, developed by Kaiser et al. (2014) within the MESSy system (Jöckel et al., 2010). To simulate the dispersion of the plume box (SP) in the background box (BG), a one-way interface is implemented based on the plume diffusion dynamics as in Petry et al. (1998). Furthermore, the plume model is capable of simulating two scenarios for a single aircraft plume, namely with and without the presence of contrail ice particles in the vortex regime and their coagulation with the aerosol particles. The contrail ice crystals are assumed to sublimate at the beginning of the dispersion regime, releasing the coagulated aerosol particles back to the SP box.

The double-box aircraft exhaust plume model has been implemented as an extension of the standard MADE3 single-box model configuration within the MESSy framework. Numerical tests have been performed to ensure the backward compatibility with the MADE3 single-box configuration of Kaiser et al. (2014). The double-box configuration has been further tested to ensure binary identical results when both boxes are initialised with the same parameters and the one-way plume dispersion is deactivated: this ensures that the core microphysical processes of MADE3 are identically represented in both boxes and that no numerical artifacts are introduced by the extension from single- to double-box model. As a result of the high flexibility of the MESSy interface, the model is fully configurable via a Fortran namelist, allowing to switch between the single-box and double-box configuration, and to run the double-box model either in the plume dispersion or in the instantaneous dispersion mode (see Sect. 2.5). The model initialization parameters (temperature, pressure, initial tracer concentrations, etc.) are also fully configurable via namelist.

## 2.2 Structure and components




Two model approaches are applied and compared to quantify the impacts on the particle properties at the end of the dispersion regime due to aviation effects. In the instantaneous dispersion approach (as shown in Fig. 1a) aviation emissions are released at time  $t_0$  and instantaneously and homogeneously dispersed in the model grid box (ID box). The evolution of the concentration  $C_i^{\rm ID}(t)$  for each species i is then simulated by MADE3, mimicking the behaviour of the global model for a single box. In the plume approach (Fig. 1b), aviation emissions are released at time  $t_0$  within the SP box and their mixing with the BG box during the plume dispersion is explicitly simulated, while also accounting for the aerosol microphysics. The plume cross-section area expands both laterally (z-axis) and vertically (y-axis) while mixing with the background air. This growth is represented by the elliptical slices along the flight track (x-axis) in Fig.1b. The expansion of the plume in the direction of the flight track (x-axis) is considered negligible. The two boxes, SP and BG, as well as the concentrations therein  $(C_i^{SP}(t))$  and  $C_i^{BG}(t)$  evolve in time including an exchange between the boxes and they experience the same microphysical processes simulated by using the same MADE3 core routine. The time integration proceeds until the plume cross-section area  $(A_{\text{plume}})$  reaches the same value of the large-scale grid-box area  $(A_{gridbox})$ , which marks the end of the dispersion regime, i.e. when the plume is completely dispersed within the background. The same time integration is applied to the ID box. In the following, we will refer to this point as the reference time and will compare the results of the two approaches at this point to estimate the impact of the plume processes on simulated aerosol number concentrations. This will be expressed as a plume correction with respect to the instantaneous dispersion approach.

Figure 1. Schematic representation of the an aircraft exhaust plume inside the background shown by the double-box model highlighting the two approaches adopted in this study: a) instantaneous dispersion approach of aircraft emissions in a large-scale grid-box (ID box); b) plume approach accounting for the dispersion of a single aircraft plume (SP box) in the background (BG box). The x-axis represents the flight trajectory. The plume gradually grows and disperses with time t, both vertically (y-axis) and laterally (z-axis). The plume growth is schematically represented by the elliptical cross-section area  $A_{\text{plume}}$  increasing at each timestep behind the aircraft, while the cross-section area  $A_{\text{gridbox}}$  of the large-scale remains constant. Here,  $t_0$  denotes the initial timestep of the simulation, set at the end of the jet regime ( $\sim$ 10 seconds behind the aircraft, not accounted for in the model), whereas  $C_i^{\text{ID}}$ ,  $C_i^{\text{SP}}$  and  $C_i^{\text{BG}}$  indicate the concentration of a species i in the three boxes, respectively. The arrow on the top symbolizes an aircraft exhaust plume with different regimes based on their time of emission.

The details of the time integration within each box are shown in Fig. 2. At each model timestep, the chemical production rate of H<sub>2</sub>SO<sub>4</sub> is calculated before calling the MADE3 microphysical scheme, which integrates the aerosol mass and number concentrations by solving the aerosol dynamics equations (see Sect. 2.2.3). The explicit calculation of the H<sub>2</sub>SO<sub>4</sub> production rate during the model time integration is an improvement introduced in this work compared to the MADE3 box-model version by Kaiser et al. (2014), who assumed a prescribed constant production rate of H<sub>2</sub>SO<sub>4</sub>. Here, we opted for a more sophisticated approach, where the production rate is calculated at each timestep from the SO<sub>2</sub> and OH concentrations (see details in Sect. 2.2.1). The online sulfate production rate calculation is relevant in the context of this study, as the H<sub>2</sub>SO<sub>4</sub> production rate affects the nucleation of sulfate particles and the growth of existing particles by condensation, hence having an impact on the aerosol properties, in particular on the number concentration. The SP box includes two further processes, that are specific to


**Figure 2.** Schematic representation of the process flow of the two approaches described in this study: the instantaneous dispersion approach (left) and the plume approach (right). Here, the rectangular boxes represent the components highlighting the key processes in the two modelling approaches. The time-loop of the dispersion regime is common to both approaches and proceeds until the cross-section area of the plume in the plume approach is equivalent to the large-scale grid-box area in the instantaneous dispersion approach. The numbers (1), (2) and (3) refers to different set of processes tracked by the tendency diagnostics implemented in the model (see Sect. 2.4).

the plume approach: the diffusion routine accounting for the mixing of the plume with the entrained background (Sect. 2.2.3) and the routine for the vortex regime, which simulates the coagulation of the aerosol particles with the ice particles during the vortex regime (upto 2 minutes behind the aircraft), representing the aerosol-ice interactions occurring within a short-lived contrail in a simplified manner (Sect. 2.2.4). We will show that this is a key process in the plume evolution and has a substantial impact on the resulting aviation-induced particle number concentration. The time integration of the dispersion regime is the same in all boxes and considers a constant time-step of 60 seconds, while the short vortex regime of the SP box uses a time resolution of 1 second, given the short duration (2 minutes) of this regime. The meteorological parameters temperature, pressure and relative humidity are kept constant during the whole simulation and are identical in the three boxes. The generated output comprises of temporally resolved aerosol mass and number concentrations for all simulated species.



The initial background concentrations of the different tracers use climatological means from the global model simulations of Righi et al. (2023) for different regions, while the aviation emissions are calculated offline based on the typical emission indices and other parameters representative of a young aircraft plume. These emissions are then used to initialise the concentration of

aviation-induced species for both the SP and the ID box, considering the initial cross-sectional area of the respective boxes. Further details about the initialization procedure are given in Sect. 2.3.

# 2.2.1 Chemical production of H<sub>2</sub>SO<sub>4</sub>

SO<sub>2</sub> is formed inside the engine combustor as a result of oxidation via OH and fuel sulfur (Lee et al., 2010). The aircraft-emitted SO<sub>2</sub> upon entering the dispersion regime undergoes oxidation with the OH radical in the downstream plume which then leads to the formation of sulfate aerosols (SO<sub>4</sub>) within seconds behind the aircraft. According to the in-situ measurements by Jurkat et al. (2011), a small fraction (a few percent) of SO<sub>2</sub> mass is converted into primary SO<sub>4</sub> during the jet regime. The remaining SO<sub>2</sub> mass remains available in the system to form H<sub>2</sub>SO<sub>4</sub>, which serve as a precursor gas for the formation of additional SO<sub>4</sub> during the dispersion regime, either via nucleation or by condensation on existing particles. Sulfate production is a slow process and therefore is calculated only in the dispersion regime (Fig. 2; blue box) and neglected in the vortex regime (Miake-Lye et al., 1993; Kärcher et al., 2007). The latter only accounts for aerosol-aerosol and aerosol-ice coagulation. The formation of H<sub>2</sub>SO<sub>4</sub> in the gas phase occurs via the third-body reaction:

$$P_{\rm H_2SO_4} = k_{\rm 3rd} \,[{\rm SO_2}] \,[{\rm OH}],$$
 (1)

where the reaction rate  $k_{3rd}$  can be calculated as:

$$k_{3\text{rd}} = \frac{k_0(T)C}{1 + k_{\text{ratio}}(T)} f_c^{\frac{1}{1 + [\log_{10}(k_{\text{ratio}}(T))]^2}},$$
 (2)

with:



$$k_{\text{ratio}}(T) = \frac{k_0(T)C}{k_{\text{inf}}(T)}; \quad k_0(T) = k_0^{300} \left(\frac{300 \text{ K}}{T}\right)^n; \quad k_{\text{inf}}(T) = k_{\text{inf}}^{300} \left(\frac{300 \text{ K}}{T}\right)^m.$$
 (3)

The parameters  $f_c$ ,  $k_0^300$ ,  $k_{\rm inf}^{300}$ , n and m are taken from the MECCA chemical scheme (Module Efficiently Calculating the Chemistry of the Atmosphere; Sander et al., 2019). The additional tracers  $SO_2$  and OH are added in the plume-model as they were not included in the original MADE3 box-model version. Note, however, that OH can only be prescribed with a constant mixing ratio in the current configuration of the plume model. A time-varying mixing ratio for OH, reproducing for example its typical daily cycle in the upper troposphere, will be considered for future versions of the model or can be considered when coupling the plume model with a global model, thus using the OH concentrations simulated by the global model itself. The sulfur budget closure of the model has been validated using the tendency diagnostics (Sect. 2.4) calculated in the model for all sulfur components, i.e.  $SO_2$ ,  $H_2SO_4$  and  $SO_4$ . As shown by Fig. S1 in the supplement,  $SO_2$  is reduced as it is converted to  $H_2SO_4$  in the gas phase, which therefore increases and subsequently decreases when forming  $SO_4$  aerosols either via nucleation or condensation.

Previous studies have shown that the aviation effect on low clouds is largely driven by sulfate aerosol particles (Righi et al., 2013; Kapadia et al., 2016). Therefore, our study primarily focuses on sulfate aerosol, with a particular attention to sulfur chemistry. Although  $NO_x$  is included in the plume model as a proxy for plume age, it is only defined as a passive tracer and  $NO_x$  chemistry is not considered. Given that it acts as a precursor for  $HNO_3$  and aerosol nitrate but has no direct impact

on particle number, this simplification is acceptable for the scope of this study. The role of organics, which in contrast could contribute to particle number via the nucleation of secondary organic aerosol (e.g. Liu and Matsui, 2022), is also not considered, due to the complexity of the involved chemistry, which could considerably increase the computational demand of the plume model. Nevertheless, as outlined in Sect. 4, the plume model is designed to a allow coupling with global models and thus take advantage of the detailed chemical scheme of a host model to account for other chemical pathways.

### 2.2.2 Aerosol microphysics based on MADE3






MADE3 is an aerosol microphysics scheme which is part of the MESSy system (Jöckel et al., 2010). MADE3 represents nine aerosol species in nine lognormal modes, given by the combination of three mixing states (soluble, insoluble, and mixed particles) in three size ranges, namely the Aitken, accumulation, and coarse mode, which are assumed to follow a lognormal size distribution with fixed standard deviation  $\sigma$  (Kaiser et al., 2014). In the following, we will use the MADE3 notation to indicate the modes, i.e. the indices k, a and c for the Aitken, accumulation and coarse mode, respectively, and the indices s, m and i for the soluble, mixed and insoluble states, respectively. Each of the 9 modes is then indicated by the combination of the indices (i.e., ks for the soluble Aitken mode). We note here that in this paper we refer to the MADE3 species black carbon (BC) as soot for consistency with the terminology of aviation-related literature, although black carbon and soot are not exactly the same (Petzold et al., 2013). The aerosol dynamics in MADE3 is calculated accounting for various microphysical processes such as coagulation, condensation of low-volatility gases acid onto existing particles, nucleation (new particle formation) and gas-to-particle partitioning. In MADE3, nucleation is calculated using the parameterisation by Vehkamäki et al. (2002), which depends on the H<sub>2</sub>SO<sub>4</sub> concentration in the model along with temperature and relative humidity. These parameters are then used to calculate the nucleate rates. The nucleation process initializes new ultrafine particles in the Aitken mode, which can rapidly grow through condensation or be removed through coagulation. For coagulation, MADE3 uses the Brownian coagulation kernel which was originally developed by Whitby et al. (1991) to perform the calculations of modal aerosol particle interactions via intramodal coagulation, for particles of the same mode, and intermodal coagulation, which produces larger size particles.

As mentioned above, the double-box configuration developed here shares the same core mechanism for aerosol microphysics (Fig. 2; orange box) as the box-model configuration and with additional plume processes, such as the plume diffusion dynamics and online calculation of sulfate production rate. The core physics of MADE3, however, remains unchanged.

MADE3 is a two-moment aerosol scheme, hence capable of calculating changes in both particle mass and number concentration. The MADE3 microphysics has been extensively evaluated in Kaiser et al. (2014) against its predecessor MADE (Lauer et al., 2005) and against the particle-resolving aerosol model PartMC-MOSAIC for idealized marine boundary layer conditions, concluding that MADE3 is suitable for global applications. MADE3 has also been evaluated by Kaiser et al. (2019) against ground-level and aircraft-based in-situ measurements in its global configuration as part of the global model EMAC (ECHAM/MESSy Atmospheric Chemistry), showing that it can reasonable reproduce the global distributions of aerosol mass and number concentrations, with a performance comparable to the one of other global aerosol models. EMAC with MADE3 has also been used in several studies focusing on aerosols and aerosol-cloud interactions, with a specific focus on the impacts of the transport sectors, including aviation (Righi et al., 2020, 2021; Beer et al., 2022; Righi et al., 2023; Beer et al., 2024).

Hence MADE3 is a well-established aerosol scheme and its microphysical core is a suitable basis for the development of the plume model presented in this work.

## 2.2.3 One-way plume interface and diffusion dynamics


A one-way interface is implemented in the double-box configuration (Fig. 2; yellow box) to account for the diffusion dynamics and for the entrainment of background air into the growing and dispersing aircraft plume, following the approach of Petry et al. (1998). In their study, the plume is described as a Gaussian plume, assuming the diffusion parameters typical for the upper troposphere (Schumann et al., 1995). The diffusion dynamics equation is solved as a Gauss function with horizontal ( $\sigma_h$ ), vertical ( $\sigma_v$ ) and shear ( $\sigma_s$ ) standard deviations given by:

$$\sigma_h^2(t) = \frac{2}{3}s^2 D_v t^3 + (2D_s + s\sigma_{0v}^2)st^2 + 2D_h t + \sigma_{0h}^2, \tag{4}$$

255 
$$\sigma_v^2(t) = 2D_v t + \sigma_{0v}^2,$$
 (5)

$$\sigma_s^2(t) = sD_v t^2 + (2D_s + s\sigma_{0v}^2)t. \tag{6}$$

These are time dependent and determine the rate of plume expansion and dispersion processes within the expanding plume. The diffusion coefficients, namely horizontal ( $D_h$ =10 m² s<sup>-1</sup>), vertical ( $D_v$ =0.3 m² s<sup>-1</sup>) and shear ( $D_s$ =1 m² s<sup>-1</sup>), and the initial horizontal ( $\sigma_{0h}$ =200 m), vertical ( $\sigma_{0v}$ =50 m) and shear ( $\sigma_{0s}$ =0) standard deviations, along with wind shear (s=0.004 s<sup>-1</sup>) are taken from Schumann et al. (1995). Eqs. (4)-(6) assume the plume to expand both laterally and vertically and to grow elliptically over time due to the strong vertical shrink of the wake formed by the wing tips during the vortex regime of the plume expansion. Based on these equations the growth of the plume cross-section area  $A_{\rm plume}(t)$  can be calculated as:

$$A_{\text{plume}}(t) = \pi c^2 \sqrt{\sigma_h^2(t)\sigma_v^2(t) - \sigma_s^2(t)},\tag{7}$$

where t is the time elapsed since the beginning of the dispersion regime and c is a free parameter determining the fraction of the initial plume incorporated in Gaussian plume. Here, as in Petry et al. (1998), a value of c=2.2 is chosen, representing 98.6% of exhaust incorporated in the plume. Based on the above equation, the solution to the diffusion equation is implemented in the plume box and applied to update the species concentrations  $C_i^{\rm SP}(t)$  after the entrainment of background air in the plume at each timestep t:

$$\hat{C}_{i}^{\mathrm{SP}}(t) = \frac{A_{\mathrm{plume}}(t) - A_{\mathrm{plume}}(t - \Delta t)}{A_{\mathrm{plume}}(t)} C_{i}^{\mathrm{BG}}(t) + \frac{A_{\mathrm{plume}}(t - \Delta t)}{A_{\mathrm{plume}}(t)} C_{i}^{\mathrm{SP}}(t)$$

$$(8)$$

where  $\hat{C}_i^{\mathrm{SP}}(t)$  is the updated value after entrainment, t is the current timestep and  $t-\Delta t$  is the previous timestep.

In the plume model, the routine implementing Eq. (8) is called right after the MADE3 microphysical core, as shown by the yellow box in Fig. 2. The time integration continues until the value of the plume cross-section area is equivalent to the cross-section area of the large-scale grid-box ( $A_{\rm plume} = A_{\rm gridbox}$ ). In the following, this final timestep denoted as reference time ( $t_{\rm ref}$ ) and is considered when comparing the results of instantaneous dispersion approach and plume approach. Considering a cross-section for a single large-scale grid-box of 300 km (horizontal) times 1 km (vertical), typical of the T42 resolution of the

Figure 3. Temporal evolution of the plume cross-section area behind the aircraft as simulated by the plume diffusion dynamics. Three diffusion scenarios are shown: medium (reference, black), slow (orange) and fast diffusion (blue). The horizontal line (grey) represents the the cross-section area of the large-scale grid-box ( $A_{\rm gridbox}$ =300 km<sup>2</sup>). The crossing point between the plume and the grid-box area marks the reference time (46 h for the reference case).

EMAC model used in previous studies on the aviation impact of aerosol (Righi et al., 2013, 2023), results in a large-scale area  $A_{\rm gridbox} = 300 \ {\rm km^2}$ . As shown in Fig. 3, for the set of diffusion parameters of Petry et al. (1998) this correspond to a reference time of about 46 h. Moreover, the plume model can simulate two additional diffusion scenarios by scaling the initial diffusion parameters ( $D_h$ ,  $D_v$ ,  $D_s$ , and s) by a factor of 0.75 and 1.5, resulting in a faster and slower plume diffusion with the reference time of 61 h and 30.5 h respectively.

# 2.2.4 Vortex regime



The vortex regime is initiated approximately 10 seconds following the release of emissions. At this stage, the emitted exhaust gets trapped inside the wake vortices which are formed when the vorticity sheet rolls up around the aircraft wing and wing-tips (Unterstrasser et al., 2014) and it can last up to a few minutes before the dispersion regime starts. During the vortex regime several dynamical processes such as chemical kinetics, turbulence, fluid dynamics occurs. The contrail formation behind the aircraft is determined by the Schmidt-Appleman criterion (Appleman, 1953; Schumann, 1996), which is based on several parameters such as aircraft altitude, engine and fuel type, temperatures and relative humidity (Kärcher, 1998; Unterstrasser et al., 2008; Paoli et al., 2011) and may vary with different regions. Studies suggest that about 85% of the contrails formed behind an aircraft are short-lived and may last up to 2-5 minutes (Gierens et al., 1999; Wolf et al., 2023).

The plume model concept outlined above accounts for the aerosol microphysical processes occurring during the dispersion regime of the plume evolution. However, the vortex regime could also be relevant for the aerosol population, especially in the case when a contrail is present during this regime, since the interactions of the aerosol particle with the ice crystals may impact the properties of the aerosol population. In our plume model, we make the simple assumption that the formation of ice particles occurs in the jet regime and that both ice crystal number concentration and size remain constant during the vortex regime. This simplified assumption is justified since the goal is to characterize the effect of the coagulation of the aerosol particles with the ice crystals in the vortex, mimicking the effect of a contrail which sublimates at the end of the vortex. In the plume approach, the vortex regime is represented as a separate process loop for the first 2 minutes of the simulations in the SP box (see Fig. 2; pink box). Here, the interaction of aerosols and ice is calculated via Brownian coagulation using the coagulation routine of the MADE3 microphysical core and implementing a passive tracer representing the ice crystals population. Given the short duration of this regime, the temporal resolution is increased to 1 second in the time integration of the vortex regime.

Mass and number concentrations of contrail ice crystals are initialised with the typical values corresponding to the end of jet regime (Bier and Burkhardt, 2022). In this study, we assume that during the vortex regime, the processes of additional ice crystal formation and growth are not accounted for. The sedimentation of ice crystals during this regime is also considered negligible: simple calculations of the ice crystals sedimentation velocity (Spichtinger and Gierens, 2009) results in values of about 0.1 cm s<sup>-1</sup> for the typical conditions at cruise altitude, which are small considering the cross-section area of the plume and the short duration (2 minutes) of the vortex regime in the model. This also means that the impact of scavenging by sedimenting ice crystal is inefficient for the removal of aerosol particles during the vortex regime and can be neglected (Unterstrasser, 2014). The ice crystals are assumed to completely sublimate at the end of the vortex regime and the residual aerosol mass is returned to the aerosol tracers of the SP box. To estimate the residual number, we assumed that every sublimating ice crystal releases a single aerosol particle, hence the residual aerosol number coincide with the assumed (constant) number of ice crystals, given that ice-ice coagulation is negligible due to the large size (~micron) of the crystals. Based on the residual aerosol mass and number, a residual diameter is then calculated for a lognormal size distribution and used to determine the target mode to assign the residuals to, consistent with the MADE3 modal structure. The residuals are assigned to the mixed (insoluble) mode if the soluble mass of the residual is larger (smaller) than 10%, and to the Aitken (accumulation) size mode if its diameter is smaller (larger) than 100 nm. The resulting aerosol mass and number concentration serve then as initial values for the further simulation of the plume evolution in the dispersion regime.

#### 2.3 Model initialization








As shown in Fig. 2, the model is initialised with typical background concentrations and with aviation emissions, considering representative values at cruise altitude of the commercial fleet. Background concentrations, required to initialise the ID box in the instantaneous dispersion approach and the BG box in the plume approach, and meteorological parameters are taken from a global simulation with the EMAC model (Righi et al., 2023). Background concentrations are provided as climatological means for all aerosol species simulated by MADE3 (see Data Availability), as well as for SO<sub>2</sub> and OH, as required by the online calculation of H<sub>2</sub>SO<sub>4</sub> chemical production in Eq. (1). Typical values for these concentrations can be found in Figs. 7a,

**Table 1.** Definition of the regions considered to initialise background conditions in this study. The regions' boundaries follow Teoh et al. (2024). The resulting parameters are from the reference simulation of Righi et al. (2023), representative of 2015 conditions. Note that the vertical selection is identical in all regions and correspond to model levels 18 and 19 of EMAC. The North Atlantic region is used as a reference, other regions are investigated as part of sensitivity studies (see Sect. 3.3).

| Region         | Latitude   | Longitude   | Temperature [K] | Pressure [hPa] | RH [%] | Number of ensembles |
|----------------|------------|-------------|-----------------|----------------|--------|---------------------|
| North Atlantic | 40-63° N   | 70-5°W      | 215-220         | 210-240        | 40-70  | 432                 |
| Europe         | 35-60° N   | 12°W-20°E   | 213-220         | 210-240        | 48-70  | 240                 |
| USA            | 23-50° N   | 126-66°W    | 214-225         | 210-240        | 49-70  | 460                 |
| China          | 18-53.5° N | 73.5-135°E  | 212-230         | 210-240        | 50-76  | 644                 |
| North Pacific  | 35-65° N   | 140°E-120°W | 214-220         | 210-240        | 49-66  | 864                 |

S13-S18 of Righi et al. (2023). The plume model simulations for this study are performed for different regions (Table 1), with the reference simulation focusing the North Atlantic region. For all regions, data from the EMAC model hybrid levels 18 and 19 are considered (corresponding to an altitude of about 10-11 km). In order to account for the spatial variability of background conditions, the climatological mean values of each of the EMAC model grid-boxes within each region are used to initialise multiple ensemble simulations of the plume model and the ensemble mean of the results is considered for the subsequent analysis.

The initial concentrations induced by aviation emissions  $C_{\text{aviation}}(t_0)$  are calculated offline using aircraft operational parameters and the following equation:

$$C_i^{\text{aviation}}(t_0) = \frac{\text{EI}_i f}{A_{\text{plume}}(t_0) v},\tag{9}$$

where  $\mathrm{EI}_i$  is the emission index of the species i, f the fuel flow (in kg s<sup>-1</sup>),  $A_{\mathrm{plume}}(t_0) = 0.15 \ \mathrm{km}^2$  is the initial plume area calculated using Eq. (7) and v the aircraft speed (in m s<sup>-1</sup>). Eq. (9) basically converts aviation emissions to concentrations by assuming that the species are released within a given initial volume, determined by the initial plume cross-section area and the aircraft speed. The values for these parameters depend on the aircraft and engine type and may be subject to a large variability. All initial parameters we consider in this study typically represent the wide-body aircraft types like Boeing 747 (Petry et al., 1998), Airbus 340 (Unterstrasser, 2014) and the DLR ATTAS (a twin-engine aircraft with medium bypass ratio turbofan). The initial size parameters for emitted primary and secondary mode particles are based on Petzold et al. (1999) which represents Rolls Royce/SNECMA M45H Mk501 turbofan of DLR ATTAS (details are shown in Table 2).



The aviation emissions are calculated for young plume conditions. In both modelling approaches (instantaneous dilution and plume), we assume that the emitted exhaust is mixed with the existing background at timestep  $t_0$ . In the plume approach, the aerosols are transformed gradually within the dispersing aircraft plume, hence in the plume model the initial concentration of the aviation emissions  $(C_i^{\text{aviation}}(t_0))$  is simply added to the background concentrations  $C_i^{\text{BG}}(t_0)$ :

$$C_i^{\text{SP}}(t_0) = C_i^{\text{aviation}}(t_0) + C_i^{\text{BG}}(t_0)$$
 (10)

The BG box concentrations initialised from the EMAC simulations undergo an initial spin-up of 6 hours to ensure internal consistency of the different tracers prior to entering the time loop.

In the instantaneous dilution approach, the ID box is initialised diluting the emissions inside a large-scale grid-box with a cross-section area  $A_{\rm gridbox}=300~{\rm km^2}$ . Therefore, the emission values are scaled to the grid-box area with respect to the initial plume area  $A_{\rm plume}(t_0)$  within the model and later mixed with the background concentrations:

$$C_i^{\text{ID}}(t_0) = \frac{A_{\text{plume}}(t_0)}{A_{\text{gridbox}}} C_i^{\text{aviation}}(t_0) + C_i^{\text{BG}}(t_0)$$
(11)

For the reference case, we use the average EI values for  $SO_2$  mass as provided in Lee et al. (2010), while the initialization of soot and ice particles is based on number EI from Bier and Burkhardt (2022). We recall that  $SO_2$  is added as a new gaseous species in the MADE3 box- and plume-model version in order to drive the online sulfur chemistry for the production of  $H_2SO_4$  (see Sect. 2.2.1). To calculate the initial  $SO_2$  mass, we assumed an EI value of  $0.8 \text{ g}(SO_2) \text{ kg}_{\text{fuel}}^{-1}$ . Thereafter, the initial  $SO_4$  mass is also calculated offline based on the primary  $SO_4$  based on initial  $SO_2$  concentration using Eq. (12) assuming the initial diameter of 2.5 nm which accounts for primary mode particles (Table 2). The initial aerosol sulfate mass  $M_{SO_4}$  is derived as a mass fraction  $\epsilon$  of emitted  $SO_2$  concentration:

$$M_{\rm SO_4} = \epsilon \, M_{\rm SO_2} \frac{\mu_{\rm SO_4}}{\mu_{\rm SO_2}}$$
 (12)

where  $\mu$  indicates the molecular weight.


Aviation emitted soot is initialised in terms of number concentration, based on the number emission index  $EI_{soot}$  by Bier and Burkhardt (2022). For the plume scenario with the availability of contrail ice particles in the vortex regime, we initialise the ice crystals as coarse mode insoluble particles with the available ice number concentration ( $N_{ice}$ ) corresponding to the  $EI_{soot}$  value from Bier and Burkhardt (2022).

To convert initialised mass concentrations to number concentrations (or vice versa) for both aerosol and ice particles, we apply the standard equation for lognormal distributions derived from the third moment:

$$N_{i,j} = \frac{6}{\pi} \frac{M_{i,j}}{D_{i,j}^3 \exp(4.5 \ln^2 \sigma_{i,j}) \rho_i}$$
(13)

where  $N_{i,j}$ ,  $M_{i,j}$  and  $\rho_{i,j}$  are the number, mass and density, respectively, for the aerosol species (or ice) i in the mode j. The value of the lognormal parameters geometric mean diameter  $D_{i,j}$  and geometric standard deviation  $\sigma_{i,j}$  are based on existing research and shown in Table 2.

#### 2.4 Tendency diagnostics



To support the interpretation of the model results and characterize them at the process level, tendency diagnostics have been implemented in the MADE3 core routines to track the changes in the tracer concentration due to each microphysical process. The tracked processes are categorized in three groups, indicated by the numbers on Fig. 2: (1) chemical processes (Sect. 2.2.1), explicitly accounting for the online sulfate production rate; (2) aerosol microphysics (Sect. 2.2.2); and (3) plume processes (Sect. 2.2.3). The tendency diagnostics are computed both in the single-box and in the double-box configuration, except for the

**Table 2.** Parameters used for the calculations of the initial aviation-induced concentrations for the reference simulation in this study. Primary sulfate emissions are assigned to the soluble Aitken mode (ks). Soot emissions are assigned to the Aitken and accumulation modes, with a 80/20% split between the insoluble and mixed modes (ki/km and ai/am). Ice is temporarily assigned to the insoluble coarse mode (ci) during the vortex regime.

| Parameter                                      | eter Description                      |                             | Reference value      | Reference(s)              |  |
|------------------------------------------------|---------------------------------------|-----------------------------|----------------------|---------------------------|--|
| f                                              | Fuel flow                             | $kg s^{-1}$                 | 0.426                | Petzold et al. (1999)     |  |
| $A_{ m plume}(t_0)$                            | Initial plume area                    | $\mathrm{m}^2$              | $1.5 \times 10^{5}$  | Petry et al. (1998)       |  |
| v                                              | Aircraft speed                        | ${\rm ms^{-1}}$             | 167                  | Petzold et al. (1999)     |  |
| $\mathrm{EI}_{\mathrm{soot}}$                  | Soot number emission index            | ${ m kg}_{ m fuel}^{-1}$    | $1.5 \times 10^{15}$ | Bier and Burkhardt (2022) |  |
| $N_{ m ice}$                                   | Ice crystal number concentration      | ${\rm cm}^{-3}$             | 222                  | Bier and Burkhardt (2022) |  |
| $\mathrm{EI}_{\mathrm{SO}_2}$                  | SO <sub>2</sub> mass emission index   | $g(SO_2)kg_{\rm fuel}^{-1}$ | 0.8                  | Lee et al. (2010)         |  |
| $\epsilon$                                     | Primary SO <sub>4</sub> mass fraction | %                           | 2.3                  | Jurkat et al. (2011)      |  |
| $\mathrm{D_{soot,k}};\sigma_{\mathrm{soot,k}}$ | Soot lognormal parameters (Aitken)    | nm; -                       | 25; 1.55             | Petzold et al. (1999)     |  |
| $\mathrm{D_{soot,a}};\sigma_{\mathrm{soot,a}}$ | Soot lognormal parameters (accum.)    | nm; -                       | 150; 1.65            | Petzold et al. (1999)     |  |
| $D_{SO_4,ks}; \sigma_{SO_4,ks}$                | SO <sub>4</sub> lognormal parameters  | nm; -                       | 2.5; 1.7             | Kärcher et al. (2007)     |  |
| $D_{ m ice,ci};\sigma_{ m ice,ci}$             | Ice lognormal parameters              | nm; -                       | 2300; 2.2            | Unterstrasser (2014)      |  |

tendencies for the plume processes, which are only available in the double-box configuration. The model provides a detailed tendency output for all species both gases and aerosol, the latter also in the different aerosol modes.

# 2.5 Implementation in the MESSy framework

The plume model is implemented in the MESSy framework as an extension of the MADE3 box-model by Kaiser et al. (2014) and its configuration can be fully controlled via a single Fortran namelist (made3.nml). This allows the user to operate it either as single-box model or plume model, and, for the latter, between the instantaneous dispersion approach and the plume approach. The namelist includes two sections: BOXINIT is used to control either the box-model configuration or the BG-box of the plume model configuration, whereas PLUMEINIT controls the SP and the ID box of the plume model configuration. The meteorological parameters, namely temperature, pressure and relative humidity, and the time parameters are initialised only once since in the BOXINIT they are the same in both boxes. A detailed list of the namelist parameters is shown in Table 3.

#### 3 Model application and results


This section explores the plume model output in terms of aviation-induced aerosol number concentrations and lognormal size distributions at the end of the dispersion regime, from both model approaches: plume and instantaneous dispersion for the reference setup (REF; Sect. 3.1) and a sensitivity on the representation of the nucleation process (NUC10; Sect. 3.2). All experiments are conducted under two plume scenarios, i.e. with and without the presence of contrail ice particles in the vortex

Table 3. List of the namelist parameters for the box-model configuration (BOXINIT) and double-box aircraft exhaust plume model configuration (PLUMEINIT). For simplicity, only the parameters which are relevant to this study are listed, for a complete listing see Kaiser et al. (2014).

| Parameter                                                   | Units             | Description                                                                                       |  |  |  |
|-------------------------------------------------------------|-------------------|---------------------------------------------------------------------------------------------------|--|--|--|
| BOXINIT:                                                    |                   |                                                                                                   |  |  |  |
| timesteps                                                   | -                 | Number of timesteps                                                                               |  |  |  |
| tmst                                                        | S                 | Timestep length                                                                                   |  |  |  |
| box_pressure                                                | Pa                | Pressure                                                                                          |  |  |  |
| box_temperature                                             | K                 | Temperature                                                                                       |  |  |  |
| box_relhum                                                  | -                 | Relative humidity                                                                                 |  |  |  |
| box_so4rat $\mu \mathrm{g}\mathrm{m}^{-3}\mathrm{s}^{-1}$   |                   | False for online calculation of H <sub>2</sub> SO <sub>4</sub> production rate (see Sect. 2.2.1). |  |  |  |
|                                                             |                   | (if True, a constant value for the rate must be provided).                                        |  |  |  |
| ${\tt box\_tracer} \qquad  {\tt mol \ mol^{-1}}$            |                   | Initial tracer concentrations for aerosol and gases                                               |  |  |  |
| l_plume_model                                               | -                 | True to activate the double-box configuration                                                     |  |  |  |
| PLUMEINIT:                                                  |                   |                                                                                                   |  |  |  |
| n_spinup                                                    | -                 | Number of timesteps for the spin-up phase                                                         |  |  |  |
| vortex_timesteps                                            | S                 | Number of timesteps for the vortex regime                                                         |  |  |  |
| vortex_tmst                                                 | -                 | Timestep length in the vortex regime                                                              |  |  |  |
| N_ice_vortex                                                | $\mathrm{m}^{-3}$ | Ice crystal number concentration in the vortex regime                                             |  |  |  |
| plume_so4rat $\mu \mathrm{g}\mathrm{m}^{-3}\mathrm{s}^{-1}$ |                   | Equivalent to box_so4rat, but for the second box.                                                 |  |  |  |
| plume_tracer mol mol <sup>-1</sup>                          |                   | Aviation-induced tracer concentrations for aerosol and gases.                                     |  |  |  |
| l_inst_disp                                                 | -                 | True (False) for the instantaneous dispersion (plume model) approach                              |  |  |  |
|                                                             |                   | (this controls the scaling of the aviation tracers, see Sect. 2.3).                               |  |  |  |

regime (indicated as Wice and NOice, respectively). For the reference setup, we consider typical background conditions of the North Atlantic region (Table 1) to initialise the plume model. The North Atlantic airspace is considered as one of the world's busiest aviation corridors, making it a prime region for studying the impact of aviation emissions on the climate (Lee et al., 2009). In Sect. 3.3 other regions are explored to understand the impact of background conditions on the plume model results. The sensitivity of the plume results to the initial parameters for aircraft emissions (see Table 2) is investigated by means of dedicated parametric studies (Sect. 3.4). Furthermore, we also analyse the impact of different microphysical processes (such as coagulation, condensation and nucleation) in all experiments based on the tendency diagnostics. An overview of all experiments conducted in this work is provided in Table 4.

**Table 4.** Overview of the experiments conducted in this work. Further details on the respective parameters are provided in Table 1, 2 and 5.

| Experiment    | Description                            | Contrail ice | Nucleation size | Region         | Varied parameter(s)              | Section |
|---------------|----------------------------------------|--------------|-----------------|----------------|----------------------------------|---------|
| REF(Wice)     | Reference                              | Yes          | 3.5 nm          | North Atlantic | _                                | 2.1     |
| REF(NOice)    | Reference                              | No           | 3.5 nm          | North Atlantic | _                                | 3.1     |
| NUC10(Wice)   | Sensitivity to nucleation size         | Yes          | 10 nm           | North Atlantic | Size of nucleated particles      |         |
| NUC10(NOice)  | Sensitivity to nucleation size         | No           | 10 nm           | North Atlantic | Size of nucleated particles      | 3.2     |
| Europe        | Backgr. conditions for Europe          | Yes          | 3.5 nm          | Europe         | -                                |         |
| USA           | Backgr. conditions for the USA         | Yes          | 3.5 nm          | USA            | _                                | 2.2     |
| China         | Backgr. conditions for China           | Yes          | 3.5 nm          | China          | _                                | 3.3     |
| North Pacific | Backgr. conditions for North Pacific   | Yes          | 3.5 nm          | North Pacific  | _                                |         |
| A1-A4         | Sensitivity to contrail ice number     | Yes          | 3.5 nm          | North Atlantic | $N_{ m ice}, { m EI}_{ m soot}$  |         |
| B1-B4         | Sensitivity to soot size               | Yes          | 3.5 nm          | North Atlantic | Soot size                        |         |
| C1-C4         | Sensitivity to sulfate size            | Yes          | 3.5 nm          | North Atlantic | SO <sub>4</sub> size             | 3.4     |
| D1-D4         | Sensitivity to primary SO <sub>4</sub> | Yes          | 3.5 nm          | North Atlantic | Primary SO <sub>4</sub> fraction |         |
| E1-E4         | Sensitivity to FSC                     | Yes          | 3.5 nm          | North Atlantic | FSC                              |         |

# 400 3.1 Reference case: North Atlantic region



As an example of application, we perform a plume model simulation with background conditions typical of the North Atlantic (see Table 1) and compare the output of two aforementioned model approaches, i.e. plume and instantaneous dispersion approach. The goal is to attain a quantitative understanding of the impact of the diffusion dynamics on the non-linear aerosol microphysical processes at the plume scale, which is not resolved by the ID box. We calculate the impact of aviation emissions on aerosol particles in terms of mass and number concentration, and of number size distribution. We compare their values in the two approaches at the end of the diffusion regime, i.e. at the reference time where the plume cross section area reaches the value of the large-scale grid-box ( $\sim$ 46 h). The aviation effect  $\mathcal E$  in the two approaches is calculated by subtracting the background concentration from the concentration in the SP and in the ID box, for the plume and instantaneous dispersion approach, respectively. In order to obtain the effect of microphysical processes at the plume scale on the concentration of i at the scale of the grid box, the aviation effect in the plume approach is scaled with the ratio of the cross-section areas:

$$\mathcal{E}_i^{\text{plume}}(t) = \left[ C_i^{\text{SP}}(t) - C_i^{\text{BG}}(t) \right] \frac{A_{\text{plume}}(t)}{A_{\text{gridbox}}}; \tag{14}$$

$$\mathcal{E}_i^{\text{inst.disp.}}(t) = C_i^{\text{ID}}(t) - C_i^{\text{BG}}(t), \tag{15}$$

for the plume and instantaneous dispersion approach, respectively. Note that at the reference time the scaling has no effect, as the two areas are identical by definition:  $A_{\text{plume}}(t = t_{\text{ref}}) = A_{\text{gridbox}}$ . Based on Eqs. (14) and (15), we define the plume correction  $\mathcal{P}_i(t)$  as the difference in the aviation effect calculated with the plume approach with respect to the instantaneous

Figure 4. Aviation effect  $(\mathcal{E})$  on the concentration of SO<sub>4</sub> in the two modelling approaches: plume (solid) and instantaneous dispersion (dashed) in the reference case with (a) and without (b) the short-lived contrail ice. The shaded area represent the vortex regime. Note the different units for time on the horizontal axis for the vortex and dispersion regime.

dispersion approach:





$$\mathcal{P}_{i}(t) = \frac{\mathcal{E}_{i}^{\text{plume}}(t) - \mathcal{E}_{i}^{\text{inst.disp.}}(t)}{\mathcal{E}_{i}^{\text{inst.disp.}}(t)}.$$
(16)

Fig. 4 shows the temporal evolution of the aviation-induced SO<sub>4</sub> mass in the two approaches, according to Eqs. (14) and (15), for the REF(Wice) and REF(NOice) scenario. SO<sub>4</sub> is controlled by the availability of SO<sub>2</sub> gas concentration in the model. Therefore, the concentration of SO<sub>2</sub> depletes as it is chemically converted into H<sub>2</sub>SO<sub>4</sub> gas via the oxidation with OH as explicitly calculated in the model (Sect. 2.2.1). As H<sub>2</sub>SO<sub>4</sub> is produced in the gas phase it eventually contributes to the aerosol SO<sub>4</sub> via nucleation and/or condensation, resulting in the almost linear growth observed in both approaches during the dispersion regime. Due to its slow chemical production (Kärcher et al., 2007), the oxidation of SO<sub>2</sub> in H<sub>2</sub>SO<sub>4</sub> is not accounted for during the vortex regime and starts at the beginning of dispersion regime at t=120 s. Hence, no changes in SO<sub>4</sub> mass occur during the vortex regime. Since the only active process during the vortex regime is coagulation, the SO<sub>4</sub> mass is conserved during this regime. At the end of the vortex regime (Sect. 2.2.4), we assume the complete sublimation of the contrail, releasing the residual aerosol particles back to the plume (SP box). As the sulfate production rate is initiated at the beginning of the dispersion regime, H<sub>2</sub>SO<sub>4</sub> becomes immediately available for nucleation and condensation. In the early phases of the dispersion regime, however, nucleation is favoured over condensation due to the low availability of condensation sinks and leads to a slight jump in SO<sub>4</sub> mass concentration in the REF(Wice) scenario. This is not the case in the REF(NOice) scenario, since no efficient reduction of aerosol number takes place during the vortex regime, leaving enough condensation sinks available at the beginning of the dispersion regime. The condensation of  $H_2SO_4$  (gas) is responsible for an approximately linear growth of the  $SO_4$  mass concentration in the dispersion regime for all three boxes, SP, BG and ID. The sulfur budget analysis shows the production of SO<sub>4</sub> mass dominantly due to the condensation of H<sub>2</sub>SO<sub>4</sub> (gas), which is responsible for an approximately linear growth of SO<sub>4</sub> mass in the dispersion regime (see Fig. S1 in the supplement). This also applies to the REF(NOice) scenario, although due to the absence of the vortex ice no difference between the two approaches can be seen here at the reference time.

**Figure 5.** As in Fig. 4, but for the aviation effect  $\mathcal{E}$  on aviation-induced particle number concentration.






A similar analysis is performed for the aviation-induced aerosol number concentrations to isolate the effect of microphysical processes from the plume dispersion (Fig. 5). Here, a clear difference between the plume and the instantaneous dispersion approach can be distinguished, in both scenarios. In the REF(Wice) scenario, the aviation-induced number concentration is considerably reduced during the vortex regime: due to their much larger size ( $\sim 1~\mu m$ ), the ice crystals effectively remove the Aitken-sized (< 10~nm) aerosol particles via coagulation within the first 2 minutes of the simulation. After contrail sublimation at the end of the vortex regime, the residual aerosol numbers are then returned to the aerosol phase (assuming one residual particle is left for each ice crystal), which can be seen as the slight increase in number concentration at t=120 s in Fig. 5a (solid line). During the dispersion regime, the number concentration is further reduced in both approaches, due to the coagulation process, which is more efficient in the plume approach. At the same time, nucleation events occur during the dispersion regime, contributing to increase the number by forming new sulfate particles from the gas phase. Given the small size of the particles assumed by the nucleation scheme of MADE3 (Vehkamäki et al., 2002), even a few nucleation events might have large contributions to the particle number. Hence, the two processes act in opposite directions to affect particle number concentrations. The analysis of the respective tendencies (Fig. 6) shows that both processes are more efficient in the SP than in the ID box due to the higher concentrations, especially in the early phase of the plume dispersion (first 8 hours), while they evolve similarly at later stages.

The difference in the aerosol number concentrations in the plume approach relative to the instantaneous dispersion approach varies during the simulation based on availability of aerosol number concentrations and the predominant processes in the plume model. After the first two hours of simulation the plume correction  $\mathcal{P}$  is about -23% in the REF(Wice) scenario, as a result of the coagulation during the vortex regime (Fig. 6a). As the simulation proceeds,  $\mathcal{P}$  reduces from about -17% after 6 hours to -12% after 12 hours. At the end of the dispersion regime, the plume correction is -15%, with a  $\pm 1$  standard deviation range of [-30; -0.7]%. In absolute terms this corresponds to -0.33 [-0.65; -0.02] cm<sup>-3</sup>.

In the REF(NOice) scenario (Fig. 6b), no evident change in aerosol number concentrations can be seen in the vortex regime and the number concentrations remain almost identical in both boxes, as the only active process in this regime is the aerosol-aerosol coagulation. As the dispersion regime begins, number concentrations is reduced in both approaches. As shown in

**Figure 6.** Tendency diagnostics of aerosol number concentration in both vortex (violet) and the dispersion regime (white) in the SP (solid) and ID box (dashed) for the two plume scenarios with (a) and without (b) contrail ice formation showing the two dominant processes coagulation (red) and nucleation (blue).

Fig. 6b, both nucleation and coagulation contribute to this reduction and, as in the REF(Wice) case, they are mostly effective in the early stages of the plume dispersion. The nucleation process has a similar tendency as in the REF(Wice) scenario, while the coagulation is more efficient, possibly due to the higher number concentration in the REF(NOice) scenario resulting from the absence of aerosol-ice coagulation during the vortex regime. The plume correction in this scenario is much lower and varies from -3.4% in the first 2 hours of the simulation to -3.3% at 6 hours and -2.7% at 12 hours, reaching a value of -4.2% [-20; 111% (-0.09 [-0.43; 0.25] cm<sup>-3</sup>) at the reference time.





These results demonstrate the overestimation of the aviation-induced aerosol number concentration by the instantaneous dispersion approach adopted by the global models as they have important implications for the calculation of the climate effect of aviation aerosol on low-clouds. Moreover, the comparison of the plume correction  $\mathcal{P}$  at the reference time between the two scenarios REF(Wice) and REF(NOice) highlights the importance of representing contrail ice in the vortex regime and their substantial impact on the aviation-induced particle number concentration, with  $\mathcal{P}(t_{\rm ref})$  increasing from -4.2% in the REF(NOice) scenario to -15% in the REF(Wice) scenario considering the impact of short-lived contrail ice. For future application of the plume model to correct for the sub-grid scale processes in global models, it is important to note that plume correction is relevant not only at the reference time, but also at earlier stages of the plume evolution.

To further characterize the aviation effect on the aviation-induced particle number concentration, we analyse it in terms of lognormal size distribution in Fig. 7, considering the number concentrations and particle dry diameters in the 9 aerosol modes of MADE3 at the reference time. As discussed in Sect. 2.3, aviation aerosol emissions are initialised with a particle size of 2.5 nm for aerosol sulfate and in two modes of 30 and 150 nm for soot. During the plume dispersion, processes such as coagulation and condensation contribute to the growth in particle size as the total number concentration reduces. The final distribution of the aviation effect shows a peak around 5-6 nm in both approaches, while the amplitude at the peak of the distribution is lower in the plume approach than in the instantaneous dispersion approach, both for the REF(Wice) and REF(NOice) scenarios respectively, consistent with the results shown in Fig. 5. The comparison between Fig. 7a and

b highlights again the effectiveness of the aerosol-ice coagulation in the REF(Wice) scenario, strongly reducing the particle number concentration during the vortex regime, with a clearly visible effect at the end of the dispersion regime. The plume correction to the aviation-induced aerosol number concentration discussed above mostly concerns the Aitken mode particles, which are predominantly comprised of sulfate aerosol particles initialised with 2.5 nm size. Another mode is visible around 30-50 nm and is due to soot particles, which are initialised around this size range, and also show a reduced concentration in the plume approach. In terms of the difference between the aviation effects in the two approaches (Fig. 7c,d) the REF(Wice) scenario is characterized by a reduction in Aitken size particles (4-5 nm) of about 0.22 cm<sup>-3</sup> (about -15%) for REF(Wice), whereas this value reduces to about 0.10 cm<sup>-3</sup> (-4.2%) in for REF(NOice). Although small in absolute terms, due to the fact that it represents the effect of a single plume, the relative difference between the two approaches is very relevant, implying that plume effects need to be considered in global models and corrected for when initializing aviation emissions in these models. Initializing emissions with an instantaneous dilution approach may otherwise lead to an overestimate of the aviation-induced particle number concentration and in turn to an overestimated impact on cloud droplet number concentration. The presence of contrails in the vortex regime makes this correction even more important.

## 3.2 Sensitivity to the nucleation process

The results discussed for the reference case demonstrated that the plume correction to the aviation-induced particle number concentration is determined by the concurrence of the microphysical processes, nucleation and coagulation, and their different effectiveness in the SP and ID box. While the coagulation process is represented by the model solving the classical equation for coagulation rates within and between each mode (Kaiser et al., 2014), the nucleation process is much more uncertain and needs to be parametrised. MADE3 uses the parametrisation by Vehkamäki et al. (2002), which calculates the nucleation rate as a function of temperature, relative humidity and H<sub>2</sub>SO<sub>4</sub> concentration. A critical free parameter in this parametrisation is the initial size of the newly nucleated particles, assumed to be 3.5 nm in diameter. In a global model study with MADE3, however, Kaiser et al. (2019) showed that assuming a larger diameter of 10 nm allows for a better model performance for aerosol number concentrations and size distributions in the free troposphere, where nucleation is the major source of ultrafine particles. Motivated by their results, we perform here an additional sensitivity test (hereafter NUC10) by repeating the above analysis with this alternative assumption. Note that the EMAC model simulation output used to initialise the plume model simulations also consider this assumption in a consistent way, i.e. a global simulation assuming 3.5 nm and 10 nm for the size of newly nucleated particles has been used to initialise the REF and NUC10 cases, respectively.

The aviation effect on the number concentration (Fig. 8) shows the same temporal evolution as in the REF case (Fig. 5): a very strong reduction in the vortex regime in the REF(Wice) scenario and a monotonic decrease during the dispersion phase. This decrease is much smoother than in the REF case, since the nucleation events in NUC10 contributes a factor of  $(10/3.5)^3 \simeq 23$  fewer particles due to the increase in their assumed size. Furthermore, as a result of the different initialization of the background, a lower number of particles is entrained in the NUC10 case, hence also the reduction during the vortex regime is relatively lower than in REF case. Nevertheless, the plume correction at the reference time is comparable to the REF case, about -13% [-77; 55]% (-0.24 [-1.45; 0.95] cm<sup>-3</sup>) and -4.2% [-9.1; 1.2]% (-0.07 [-0.17; 0.02] cm<sup>-3</sup>), in the NUC10(Wice)

Figure 7. Aviation effect  $\mathcal{E}$  at the reference time on the lognormal size distribution calculated from aerosol number concentrations and dry diameters at the reference time in the two approaches (plume and instantaneous dispersion) and for the two scenarios REF(Wice) (a) and REF(NOice) (b). Panels (c) and (d) show the difference between the aviation effects in the two approaches.

Figure 8. As in Fig. 5, but with the alternative assumption of 10 nm for the size of newly nucleated particles.

Figure 9. As in Fig. 6, but with the alternative assumption of 10 nm for the size of newly nucleated particles.

and NUC10(NOice) scenarios. The plume correction values in NUC10 scenarios are close to those of the REF case, but are characterized by a much larger variability, particularly in the Wice case. The tendency analysis (Fig. 9) shows an overall reduction in the efficiency of both microphysical processes in the NUC10 case as compared to the REF case. The aviation effect on the size distribution at the reference time peaks at a larger size of 10-15 nm for both scenarios compared to the REF case that peaks at 5-6 nm ( Fig. S2a,b and Fig. 7a,b). This is due to the increased size of newly nucleated particle, which corresponds to the particles with relatively larger size ( $\sim$ 20 nm) surviving towards the end of the dispersion regime. The maximum of the distribution in the instantaneous dispersion approach remains higher than in the plume approach for both scenarios (with and without ice), especially for the particles around 10-20 nm size, which are mostly SO<sub>4</sub> particles. The difference between the aviation effects in two approaches for the size distribution (Fig. S2c,d) further shows that the instantaneous dispersion approach overestimates the survival of the Aitken mode particles between at 8-10 nm and underestimates the 20 nm particles, which results from the less efficient coagulation process in the instantaneous dispersion approach as compared to the plume approach (Fig. 9).

Although the size of newly nucleated particle is an important parameter in the model, only minimal differences in the plume correction at the reference time between the REF and the NUC10 case are found.

#### 3.3 Sensitivity to background conditions





The previous sections showed an example of the application of the plume model for typical conditions over the North Atlantic, but these conditions may of course vary over other regions. This is especially important when considering background concentrations and the way they influence the aerosol microphysical processes in the plume. Nucleation events, for instance, are favoured in cleaner backgrounds, due to the lower availability of aerosol particles serving as condensation sinks. Given the key importance of the nucleation processes for the plume correction demonstrated above, it is important to analyse the plume results in different regions. Here, we consider four regions in the Northern Hemisphere, characterized by different background properties than the North Atlantic: Europe, USA, China and North Pacific (see Table 1 for details). The plume simulations are

**Figure 10.** Aviation effect on aerosol number concentration in the four regions (a) Europe, (b) USA, (c) China and (d) North Pacific. All simulations are performed based on the Wice scenarios, i.e. considering a contrail in the vortex regime.

performed using the same parameters of the REF case under the Wice scenario. Only the initial background concentrations  $C_i^{\mathrm{BG}}(t_0)$  are varied.




Comparing the aviation effect on particle number concentrations for the different regions in Fig. 10, we clearly observe a large variation. The plume correction at the reference time is reduced to -12% [-14; -9.4]% (-0.27 [-0.34; -0.21] cm<sup>-3</sup>) over Europe, while it is larger than for the North Atlantic over all other regions, ranging from -29% [-28; -30]% (-0.69 [-0.61; -0.76] cm<sup>-3</sup>) over USA, -42% [-66; -28]% (-0.69 [-0.82; -0.56] cm<sup>-3</sup>) over China, to about -39% [-58; -22]% (-0.73 [-1; -0.43] cm<sup>-3</sup>) over North Pacific. As all the other parameters are not varied, the resulting variability in the aviation effect of aerosol number concentrations is due to the different background conditions which eventually affect the nucleation and coagulation processes (this is confirmed by the tendency analysis in Fig. S3 and S4). Especially in the polluted background conditions, the coagulation process tends to effectively reduce the aerosol number concentration, however, the competition between nucleation and coagulation is ubiquitous. Given the sporadic behaviour of nucleation, the aerosol numbers are substantially affected especially during the first few hours of the simulation. In addition to this, we also see a rather enhanced condensation tendency in the SO<sub>4</sub> mass over the highly polluted regions such as China and USA (Fig. S4): this might lead to an efficient depletion of H<sub>2</sub>SO<sub>4</sub> at the expenses of nucleation, thus reducing the particle number with respect to other regions.

**Figure 11.** Aviation effect at the reference time in terms of lognormal size distribution for the four regions (a-d) and the respective differences between the aviation effects in two approaches (e-h).

In terms of size distribution of aviation-induced particles (Fig. 11), Europe shows a peak at around 5 nm, similar to the North Atlantic case (Fig. 7a), while this shifts to larger particles, around 7 nm, in both USA and China, which supports the hypothesis of an enhanced condensation process leading to particle growth, while partly suppressing new particle formation via nucleation. In Europe and USA, the size distributions retain the same shape in both approaches, with a lower amplitude in the plume approach as for the North Atlantic case. Over China, however, we observe a bimodal size distribution with two peaks in the plume approach at 6 nm and 30 nm, which provides the possibility of the survival for a broad range of particles towards the end of the dispersion regime. This is possibly due to the enhanced coagulation and condensation processes (see Figs. S3 and S4 in the supplement) in the polluted background conditions. Additionally, we observe the similar bimodal distribution in the North Pacific region with two peaks around 5 nm and 20 nm, possibly due to the dominance of nucleation and condensation, where the formation of new particles via nucleation is the most favoured processes in cleaner backgrounds. At the later stage however, these particles are effectively reduced by coagulation processes in the plume approach (Fig. 11d).

These results confirm that the instantaneous dispersion approach overestimates the aviation-induced particle number concentrations in all investigated regions, but the plume correction varies considerably across the regions. The properties of the aerosol population at the end of the dispersion regime are also very diverse, which highlight the importance of properly accounting for different background conditions (Fig. S5) when simulating the impact of aviation emissions on the aerosol number concentration. This needs to be taken into account for future application of the plume model in the context of global model studies of the aviation-aerosol indirect effects and for the development of parametrisations to account for the subgrid-scale aerosol processes in the aircraft plumes.

**Table 5.** List of parameters shortlisted for the parametric study. Each set of variations of a given parameter is identified by an index (A-E) and the different values of the parameters by a number (1-4) in addition to the reference (REF).

| Set | Parameter                                               | Units                              | 1          | 2        | REF       | 4        | 5          | Sources                  |  |
|-----|---------------------------------------------------------|------------------------------------|------------|----------|-----------|----------|------------|--------------------------|--|
| _   | $N_{ m ice}$                                            | $\mathrm{cm}^{-3}$                 | 90         | 163      | 222       | 268      | 350        | D:                       |  |
| A   | ${ m EI}_{ m soot}/10^{15}$                             | $\mathrm{kg}_{\mathrm{fuel}}^{-1}$ | 0.5        | 1        | 1.5       | 2        | 3          | Bier and Burkhardt (2022 |  |
| В   | $D_{\mathrm{soot,k}};\sigma_{\mathrm{soot,k}}$          | nm; -                              | 26.3; 1.68 | 28; 1.68 | 25; 1.55  | 27; 1.63 | 32.5; 1.71 | Moore et al. (2017)      |  |
| Б   | $\mathrm{D}_{\mathrm{soot,a}};\sigma_{\mathrm{soot,a}}$ | nm; -                              | -          | -        | 150; 1.65 | -        | -          | Wioofe et al. (2017)     |  |
| C   | $D_{\mathrm{SO_4,ks}};\sigma_{\mathrm{SO_4,ks}}$        | nm; -                              | 2; 1.7     | 2.2; 1.7 | 2.5; 1.7  | 2.7; 1.7 | 3; 1.7     | Kärcher et al. (2007)    |  |
| D   | $\epsilon$                                              | %                                  | 1.2        | 1.9      | 2.3       | 2.6      | 2.8        | Jurkat et al. (2011)     |  |
| E   | $\mathrm{EI}_{\mathrm{SO2}}$                            | $g(SO_2) \; kg_{\rm fuel}^{-1}$    | 0.2        | 0.4      | 0.8       | 0.6      | 1          | Lee et al. (2010)        |  |

# 3.4 Sensitivity to the aviation emission parameters




As described in Sect. 2.3 and summarized in Table 2, the initialization of the plume model depends on the characteristics of aircraft emissions. Those parameters largely depend on the aircraft operation, aircraft and engine efficiency, combustion technology and fuel characteristics. They determine the emission indices of emitted components, initial particle size in the young exhaust plume and, in the short-lived contrail scenario, also on the properties of the ice crystals. In this section, we explore the model sensitivity towards those parameters, in order to identify the most sensitive parameters with respect to the aviation effects discussed in Sect. 3.1-3.3, thus also providing insights for future measurement campaigns targeting aviation effects on aerosol and clouds (see, e.g., Voigt et al., 2017). Table 5 shows a detailed list of the parameters chosen for this parameter study and their tested values together with their respective literature references. We test the impact of each individual parameter by altering one parameter at a time, only, while keeping the others at their reference value. The reference setup is the REF simulation for the North Atlantic discussed in Sect. 3.1, under the Wice scenario.

Study A addresses the initial assumptions on number of ice crystals in a short-lived contrail ( $N_{\rm ice}$ ) and the corresponding soot number emission index EI $_{\rm soot}$  based on the simulations by Bier and Burkhardt (2022) with the ECHAM-CCMod model at 240 hPa. Study B involves different measurements during the ACCESS flight campaign on the size distributions parameters of emitted soot particles (Moore et al., 2017). Here, we consider the measurement performed with the HEFA 50:50 fuel blend, at medium thrust (simulation B1) and high thrust (B2), and with the standard Jet-A fuel, at medium thrust (B3) and high thrust (B4). Study C target the initial size of aerosol sulfate particles: no measurements are available for this parameter, but theoretical studies showed that at 10 s behind the aircraft the particles exist in the size range of 1 nm and they are mostly comprised of molecular clusters of organics. These particles are either consumed or scavenged through Brownian coagulation by the ions of the size range 2-3 nm, which are presumably the SO4 particles (Kärcher et al., 1996, 2000, 2007). Hence, we vary the sulfate size within this range for study C. The fraction  $\epsilon$  of SO2 mass converted into aerosol sulfate (primary SO4) is explored in study D, based on the measurements on different aircraft and engine types during the CONCERT campaign (Jurkat et al., 2011).

Figure 12. Plume correction  $\mathcal{P}_N$  to the aviation-induced particle number concentration for the parametric studies discussed in Sect. 3.4. The horizontal axis represent the REF and the four variations, the vertical axis the varied parameters in the respective parametric studies (see Table 5). The REF column exhibits the reference value discussed in Section 3.1. FSC stands for fuel sulfur content.

Finally, the emission index of  $SO_2$  (i.e., the fuel sulfur content) is varied based on the range of values provided by Lee et al. (2010), which are representative of the fleet average.

The results of these parametric studies are summarized in the heat map in Fig. 12 in term of plume correction  $\mathcal{P}_N$ , i.e. the relative difference on the aviation-induced particle number concentration between the plume and the instantaneous dispersion approach. The REF case result in a -15% correction, as already discussed in Sect. 3.1. The largest variability is found for the  $N_{\rm ice}$  (EI<sub>soot</sub>) parameters, with the plume correction decreasing (in absolute term) as the assumed number concentration of ice crystals in the vortex regime decreases, and for the emission index of SO<sub>2</sub>, with the plume correction being larger when assuming a low fuel sulfur content of the aviation fuel. The plume correction shows only a minimal sensitivity to the other sulfate-related parameters and to the soot size.

Study A shows an almost linear relationship between the assumed ice crystal number concentration in the vortex regime ( $N_{\rm ice}$  and the associated variation of  $EI_{\rm soot}$ ) and the plume correction for aviation-induced number concentration, which varies between -9.4% [-25; 5.6]% (-0.21 [-0.54; 0.13] cm $^{-3}$ ) for  $N_{\rm ice}$ =90 cm $^{-3}$  and -21% [-35; -7.0]% (-0.45 [-0.74; -0.15] cm $^{-3}$ ) for  $N_{\rm ice}$ =350 cm $^{-3}$ . This is due to the aerosol-ice coagulation in the vortex regime, increasing its efficiency as the number of

ice crystals increases (see Fig. S6, study A), thus allowing for a more efficient reduction of aerosol particles, and confirms the importance of the vortex regime for the plume effects already highlighted in Sect. 3.1. The size distributions (Fig. 13a,b) further show the strong sensitivity the aviation-induced number concentration to the number of ice crystals in the plume approach, while no sensitivity is seen in the instantaneous dispersion approach, which does not include a vortex regime.







The impact on the results of the assumptions on the soot size distribution parameters (study B) shows very small variations in the plume correction. The size distributions (Fig. 13c,d) show basically no substantial changes across the different simulations. On the one hand, this is due to the consistency of the in situ measurements on the soot size distribution parameters (Petzold et al., 1999; Moore et al., 2017), which generally agree on a diameter of about 25-30 nm, with a weak dependency on the fuel type, as recently confirmed by the large scale and long term measurements from the In-service Aircraft for a Global Observing System (IAGOS) experiment (Mahnke et al., 2024). On the other hand, study B also reveals that the role of soot on the aviation-induced particle number concentration is marginal, as this is mostly affected by the smaller, nanometer size, sulfate aerosol particles. The impact of soot size is therefore negligible in terms of plume correction as well as aviation effect (Fig. S6, study B).

The variation of the initial sulfate size (study C) has very little impact on the plume corrections. This is quite surprising, given the overwhelming importance of sulfate for the aviation number concentration. A decrease in the initial sulfate size leads to only a slight increase of the aviation effect in both approaches and to a slightly more negative plume correction, from –15% [–30; –0.67]% (–0.33 [–0.64; –0.01] cm<sup>-3</sup>) to –16% [–31; –1.2]% (–0.35 [–0.68; –0.03] cm<sup>-3</sup>) when reducing the sulfate size from 3 nm to 2 nm (Fig. 13e,f). According to the tendency analysis performed on the aerosol number concentration for study C, this can be explained with the coagulation process increasing its effectiveness as the initial size is reduced and hence more sulfate particles are emitted (see Fig. S7 in the supplement). This is particularly the case during the vortex regime, where the coagulation-driven reduction in the aviation effect becomes stronger as the initial number increases. At the end of the dispersion process, the plume correction converges towards a similar value, regardless of the initial number concentration of sulfate particles.

A similar result is also obtained in study D for the variation of the primary  $SO_4$  fraction  $\epsilon$ , with the plume correction remaining almost constant at the value of the REF experiment for the whole range of tested values of this parameter and no changes in the size distributions (Fig. 13g,h). The reason is similar as for study C: an increase in the primary  $SO_4$  fraction results in an increase of the emitted mass and hence number of emitted number of particles. The increase in number concentration is then again compensated by a more effective coagulation process, predominantly in the vortex regime. A further reason could be that increasing the primary  $SO_4$  fraction slightly reduces the availability of  $SO_2$  and eventually  $H_2SO_4$ , thus reducing the impact of nucleation on particle number. This could explain the smaller variability of study D compared to study C, although both have a similar impact on the initial particle number concentration.

The plume model is highly sensitive towards the emission index of  $SO_2$  as shown in study E (Fig. 12). The plume correction shows a large variation across the range of reported literature values for this parameters, from -62% [-125; -2.0]% (-0.32 [-0.63; -0.01] cm<sup>-3</sup>) for a low fuel sulfur content of 0.2 g( $SO_2$ ) kg<sup>-1</sup> to -12% [-24; -0.8]% (-0.34 [-0.66; -0.02] cm<sup>-3</sup>) for a high fuel sulfur content of 1 g( $SO_2$ ) kg<sup>-1</sup>. This large variation in the relative plume correction is related to the strong variation

**Figure 13.** As in Fig. 7, aviation effect in terms of lognormal size distribution at the reference time (left) and the respective difference between the aviation effect in two approaches (right) for the parametric studies discussed in Sect. 3.4.

in the aviation effect increases in both approaches, while the absolute difference between the aviation effects in two approaches remains fairly constant (Fig. 13g,h). As the fuel sulfur content controls both the  $SO_2$  and, via the primary  $SO_4$  fraction  $\epsilon$ , the  $SO_4$  initial concentrations in the model, the nucleation tendency gains importance as more  $SO_2$  and eventually  $H_2SO_4$  becomes available with increasing fuel sulfur content, although the number tendency is still controlled by coagulation (Fig. S7), whereas the  $SO_4$  mass is predominantly controlled by condensation process (Fig. S8).

# 4 Discussion







The goal of the development of the plume model presented in this work is to address the aerosol microphysical processes at the plume scale, while following the plume expansion and mixing with the background air, thus improving over the simple instantaneous dispersion approach adopted by global models. Given that the targeted application scope of the plume model, its development has focused on the aerosol microphysics and on specific aspects of the plume dynamics, whereas other processes have been addressed in a simplified way, resulting in limitations and uncertainties that need to be considered when applying the plume model results for application studies:

- plume dynamics: complex plume dynamic processes such as turbulence, processes inside primary and secondary wakes,
   contrail formation, evolution and decay of contrails are beyond the capabilities of the model presented here. Dedicated models (such as Unterstrasser, 2014) are available which explicitly simulate these complex dynamic processes.
- short-lived contrail ice: the representation of short-lived contrail ice in our model is very simplified, as the goal is to represent ice crystals as a coagulation sink for aerosol particles and to estimate the reduction in aerosol number concentration during the vortex regime. To this purpose, a simple scenario mimicking a short-lived contrail represented by a passive tracer for ice crystals with a constant number concentration was considered. More realistic representations are of course possible, but the increased complexity would likely have a limited impact on the main conclusions of this study and might make future implementations in global models more challenging. For global model applications, however, the role of persistent contrails should also be considered, as they may further enhance the reduction of aviation-induced particle number concentrations under specific atmospheric conditions (Schumann, 1996).
- constant meteorology: parameters such as temperature, pressure and relative humidity are initialised as a constant in the model. Provided the very short duration of the vortex regime (2 minutes), this is well justified in this regime. For the dispersion regime, Schumann et al. (1995) showed that the vertical diffusivity is negligible compared to the horizontal one. Hence, we consider that it is reasonable to assume no temperature gradient along the plume and assume constant meteorological parameters for the plume. This assumption is also justified in view of future applications in global models.
- sulfate production rate: the third-body reaction in Eq. (1) assumes a constant OH concentration and we prescribed an
  average value representative for the upper troposphere. This is of course a simplification, as OH is a short-lived compound
  characterized by a strong temporal and spatial variability. Since the plume model does not account for detailed gas phase

chemistry, this is a necessary simplification. It can, however, for example by coupling the plume model with a global model, which can provide this information via its chemical scheme.

- modal aerosol scheme: MADE3 is a modal scheme, originally developed for regional models and later extended for global applications (Kaiser et al., 2014). When applying it to the plume scale, the (strong) assumption that the physical and chemical processes can still be reasonably represented must be acknowledged. Particle-resolving models may be more suitable to track aerosol processes at the plume scale, but in view of future applications of the plume model for global model studies this is not a feasible option, due to the large computational demand of particle-resolving models.







Given these limitations, a direct evaluation of the plume model result against observational data is challenging, also due to the fact that direct measurements in the very early plume stages are difficult to achieve due to safety constraints and instrumentation limitations, especially in the presence of contrail ice. The model output for the aged plume remains suitable for evaluation and can be compared against in-situ data from aircraft campaigns. While such measurements are inherently limited in spatial coverage and plume positioning, CO<sub>2</sub> and NO<sub>x</sub> concentrations can be used as proxies for plume age, and our model is able to provide consistent plume age estimates based on these tracers, as shown in Mahnke et al. (2024). A further option would be to compare against large-scale in situ measurements as provided, for instance, by IAGOS (Brenninkmeijer et al., 2007): this, however, would require extending the model to account for multiple plumes, and for their interaction and superposition across various geometries. Besides evaluation purposes, future in situ measurements targeting aircraft plumes could also provide valuable data for model initialisation, especially concerning the size-resolved properties of volatile nucleation mode particles (Dischl et al., 2024; Williamson et al., 2018).

The plume model presented here is ready for application and can be coupled with global models, to calculate the changes in specific parameters of the aviation aerosol population such as mass, number and size, between the time of emission (i.e., the end of jet regime) and the end of the dispersion regime. This can provide the global models with refined and more physically motivated assumptions to initialise aviation emissions in global simulations, considering the plume-scale processes that these models cannot resolve. The aviation emission inventories required to initialise global aerosol-climate models for assessing the climate impact of aviation usually provide the mass of emitted species, but do not include particle number emissions, especially for volatile particles that are relevant for the aviation impact on low clouds. Converting mass to particle number requires assumptions about the particle size distribution at emission. Current methods are hampered by the scarcity of observational data at different plume stages, especially on particle number and size, which introduces uncertainty and potential biases in representing aviation-induced aerosols. This challenge is compounded by the fact that particle size distributions evolve over time, with younger plumes typically containing larger concentrations of smaller-sized particles, but global models cannot resolve this evolution due to their coarse resolution. Our plume model could also suffer from the scarcity of available observational data for initialisation and shows high sensitivity to some of these initial parameters, but it effectively helps to bridge the gap between the beginning of the vortex regime and the large scale of global models, producing detailed output in terms of species mass, and number concentrations categorized by size mode and mixing states at different stages of the plume evolution. From these results, number-to-mass ratios can be derived for each aerosol mode and used to calculate more physically based emitted particle numbers. Given the low computational costs of the plume model, it can also be employed as an online parametrisation, thereby more accurately accounting for the impact of local conditions and detailed background chemistry.

The negative plume corrections quantified in this work directly impact the aviation-induced particle number concentration and, when considered in global models, may impact the aviation-induced perturbations to cloud droplet number concentration and hence the climate effect from aerosol-cloud interactions (ERFaci) in low-level clouds. This, however, also depends on the assumptions made so far by global models to implicitly account for plume effects (e.g., assuming larger particle size upon emissions). Nevertheless, previous studies (e.g., Gettelman et al., 2013; Righi et al., 2013) demonstrated the strong sensitivity of ERFaci in low clouds to the size distribution assumptions for emitted particles, leading to a range of climate forcing estimates spanning several tens of mW m<sup>-2</sup>. A parametrization based on the plume model will help better constraining this uncertainty, thus resulting in more precise estimates, supporting more robust future emission policies. Thanks to its flexibility, the plume model can also be easily extended with additional parameters for evaluating aerosol emissions from future aircraft technologies (e.g., lean-burn and rich-burn engines) and alternative fuels such as sustainable aviation fuel (SAF), offering critical insights for sustainable aviation development. In low-sulfur scenarios connected with SAF usage, ERFaci on low clouds may become less relevant, which makes a robust quantification under current fleet conditions even more important in view of a possible loss of cooling due to low-sulfur fuels, similar to the current debate on the climate impact of low-sulfur regulations in shipping (Yoshioka et al., 2024). While here we assumed engine and fuel parameters mostly characteristic of conventional Jet-A1 combustion, future shifts in fuel composition or engine design may influence the ice nucleation efficiency in contrails and the microphysics of aerosol-cloud interactions. Likewise, changes in traffic patterns through trajectory optimization or rerouting could amplify the role of microphysical processes, underscoring the importance of plume-scale approaches in future aviation-climate assessments.

## 5 Conclusions and outlook





In this study, an aircraft exhaust plume model based on the MADE3 aerosol microphysical scheme has been developed to simulate the subgrid-scale microphysical processes of aviation-induced aerosol particles inside an expanding and dispersing aircraft plume within the background at typical cruise levels. The plume model has been developed by extending the MADE3 box model configuration to a double box configuration representing the plume and the background, respectively, with a one-way interface to simulate the dispersion of the plume into the background and a simplified representation of the vortex regime. The latter mimics the formation of a short-lived contrail in the plume and allows to estimate the effect of the interactions between aerosol particles and ice crystals via coagulation.

We compared the plume approach with an instantaneous dispersion approach, in which aviation emissions are instantaneously dispersed over large spatial scales as commonly done in global models. Based on these two approaches, a plume correction was quantified to characterize the effect of the plume scale processes on aviation-induced particle number concentration at the end of the dispersion phase of the plume.

Several plume model simulations have been performed to address the model sensitivity to different parameters and assumptions. The main conclusions from these experiments can be summarized as follows:

1. For typical cruise conditions over the North Atlantic, the plume model simulates an aviation-induced particle number concentration at the end of the plume dispersion (plume correction) 15% lower than the instantaneous dispersion approach.



- 2. The bulk of this reduction in aerosol number concentration is due to the coagulation of aerosol particle with the (larger) ice crystals during the vortex regime. An alternative scenario without short-lived contrail ice particles resulted in a much smaller plume correction of -4.2%.
- 3. Compared to the instantaneous dispersion approach, the effect of plume processes on aerosol sulfate and soot mass was found to be negligible, as the total mass is conserved. However, minor differences in the condensation and nucleation rates in the two approaches were found, affecting the partitioning between the gas and aerosol phase.
- 4. A detailed process-level analysis showed that coagulation and nucleation are identified as the most relevant processes controlling the aviation-induced particle number concentration during the plume dispersion regime and the corresponding plume correction with respect to the instantaneous dispersion approach. A sensitivity study shows that reducing the efficiency of nucleation is compensated by a reduced coagulation, so that the plume correction at the end of the dispersion phase is only weakly affected.
- 5. The background concentrations of aerosol and precursor gases used to initialise the plume model have a considerable impact on the results. Simulations performed over different regions of the Northern Hemisphere show large variations in the plume correction, from –12% over Europe to –42% over China.
  - 6. Parametric studies on several aviation emission parameters used for model initialization show a strong sensitivity of the plume correction to the assumed ice crystal number concentration in the scenario with a contrail in the vortex regime (controlling the efficiency of the nucleation with aerosol particles and thus their number reduction) and to the fuel sulfur content (impacting the SO<sub>2</sub> emission and consequently, the formation of H<sub>2</sub>SO<sub>4</sub>, eventually leading to new particle formation via nucleation). Despite the dominant importance of aviation-induced sulfate particles over soot particles, the model shows a weak sensitivity to sulfate-related parameters, such as the amount and size of the primary SO<sub>4</sub> particle fraction.

Overall, the plume model developed and applied in this study demonstrated that the aerosol microphysical processes, in particular coagulation and nucleation, are more efficient at the plume scale compared to the large-scale grid-box approach of global models, and that accounting for this sub-grid scale effects leads to a substantial reduction in the aviation-induced aerosol number concentrations at the end of the dispersion regime of an aircraft plume. Thanks to its flexibility and its very low computational demand, the plume model is suitable for both offline and online parametrisations of the aviation-induced particle number concentrations in the early stages after the emissions. The many sensitivity simulations performed in this study

will help to narrow the number of parameters to be considered when applying the model for global applications, also providing hints to future measurement campaigns targeting these effects. The latest campaign datasets such as the one from ECLIF 3 (Dischl et al., 2024; Märkl et al., 2024), can be utilized to initialise the plume model for future simulations. While the current version of the plume model addresses only a single aircraft plume, future adaptations could enable the examination of scenarios involving multiple and/or partially overlapping plumes. Future estimates of the aviation-induced climate effects via aerosol-cloud interactions may benefit from this model to include more robust assumptions on the properties of aviation aerosol upon and shortly after emissions, with important consequences for the simulations of the aerosol life cycle and its interactions with clouds.

Code availability. The Modular Earth Submodel System (MESSy, https://doi.org/10.5281/zenodo.8360186; The MESSy Consortium, 2024) is being continuously developed and applied by a consortium of institutions. The usage of MESSy and access to the source code are licensed to all affiliates of institutions which are members of the MESSy Consortium. Institutions can become a member of the MESSy Consortium by signing the MESSy Memorandum of Understanding. More information can be found on the MESSy Consortium Website (http://www.messy-interface.org, last access: 10 July 2024). The code presented and used here (https://doi.org/10.5281/zenodo.13134188) has been based on MESSy version 2.55.2 (https://doi.org/10.5281/zenodo.13134188) and will be part of the next official release.

Data availability. The Fortran namelists and the data to initialise the double-box aircraft exhaust plume model, and the output of the simulations analysed in this study are available at https://doi.org/10.5281/zenodo.17192582 (Sharma, 2025).

Author contributions. MS developed the model, performed the simulations, analysed the results and wrote the paper. MR contributed to the concept, to the model development and analysis. JH, AS, DS and VG contributed to the concept and to the interpretation of the results. All co-authors contributed to the writing.

Competing interests. One of the co-authors (Volker Grewe) is topical editor for Geosci. Model Devel.

790 Acknowledgements. We are grateful to Ulrike Burkhardt, Klaus Gierens, Christopher Kaiser, Bernd Kärcher and Christiane Voigt (DLR), Christoph Mahnke and Andreas Petzold (FZJ), and Holger Tost (JGU Mainz) for helpful discussions. Simon Unterstrasser (DLR) provided useful comments on an earlier version of this manuscript. The model simulations and data analysis for this work used the resources of the Deutsches Klimarechenzentrum (DKRZ) granted by its Scientific Steering Committee (WLA) under project IDs bd0080 and bb1393. This study was supported by the DLR aviation research programme (Eco2Fly project) and by the European Commission via their Horizon 2020 Research and Innovation Programme (ACACIA project, grant no. 875036).

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
