# Peer review of "A double-box model for aircraft exhaust plumes based on the MADE3 aerosol microphysics (MADE3 v4.0)"

_EGUsphere, 2025_

## Author Comment (AC3)

We are grateful to both reviewers for their insightful comments and suggestions, which helped us to improve the manuscript. A detailed point-by-point reply to each comment can be found below: reviewers' comments are shown in black; our reply is shown in blue and text passages from the paper are shown in *italic red*.

**Referee 1**

The study briefly mentions that the difference between aerosol number concentrations between the instantaneous dispersion and plume approaches can have "significant effects implications for the calculation of the climate effect of aviation aerosol on low clouds".

I would however encourage a somewhat more detailed discussion of what that effect might look like, referencing different assumptions in literature estimates and relating to the sensitivity of results shown here, e.g. in what direction this could push estimates of the ERFaci, implications for studies exploring future aviation fuels or propulsion systems (e.g. given dependence on SO2 emissions, limitation that this study is initialized with contrail ice typical for jet fuel), potential changing geographic traffic pattern etc. This would strengthen the motivation and broader relevance of the study.

We appreciate the opportunity to further elaborate on the implications of the plume correction for aerosol number concentrations and the resulting climate effect, particularly related to the aviation-induced aerosol-cloud interactions. In response, we have extended the discussion in Sect. 4 as follows:

*"The negative plume corrections quantified in this work directly impact the aviation-induced particle number concentration and, when considered in global models, may impact the aviation-induced perturbations to cloud droplet number concentration and hence the climate effect from aerosol-cloud interactions (ERFaci) in low-level clouds. This, however, also depends on the assumptions made so far by global models to implicitly account for plume effects (e.g., assuming larger particle size upon emissions). Nevertheless, previous studies (e.g., Gettelman et al., 2013; Righi et al., 2013) demonstrated the strong sensitivity of ERFaci in low clouds to the size distribution assumptions for emitted particles, leading to a range of climate forcing estimates spanning several tens of mW m$^{-2}$. A parametrization based on the plume model will help better constraining this uncertainty, thus resulting in more precise estimates, supporting more robust future emission policies. Thanks to its flexibility, the plume model can also be easily extended with additional parameters for evaluating aerosol emissions from future aircraft technologies (e.g., lean-burn and rich-burn engines) and alternative fuels such as sustainable aviation fuel (SAF), offering critical insights for sustainable aviation development. In low-sulfur scenarios connected with SAF usage, ERFaci on low clouds may become less relevant, which makes a robust quantification under current fleet conditions even more important in view of a possible loss of cooling due to low-sulfur fuels, similar to the current debate on the climate impact of low-sulfur regulations in shipping (Yoshioka et al., 2024). While here we assumed engine and fuel parameters mostly characteristic of conventional Jet-A1 combustion, future shifts in fuel composition or engine design may influence the ice nucleation efficiency in contrails and the microphysics of aerosol-cloud interactions. Likewise, changes in traffic patterns through trajectory*

*optimization or rerouting could amplify the role of microphysical processes, underscoring the importance of plume-scale approaches in future aviation-climate assessments."*

We would, however, refrain from speculating about how the climate effect could look like, since a quantitative estimate is challenging for several reasons. The effect of downward transport of aerosol particles, for instance, cannot be quantified within the current framework of the model. Moreover, as stated in the text above, the sign of the correction for ERFaci is highly sensitive to the assumptions on sulfate size in existing modelling studies, which introduces additional uncertainties. Any resulting estimate would risk being extrapolated beyond the context of our model setup and assumptions, and could therefore be misleading.

There are quite a few experiments performed. While they are described in each sub-section of the Results, I think it could be helpful for the reader to rather have a separate section for experiments under Methods, including a table with the experiment names

We have indeed performed many experiments for this study. To guide the reader through all these experiments and the respective discussion, we added a table (Table 4) at the beginning of Sect. 3 briefly summarizing their setup and pointing to the section where they are discussed.

Some figures, e.g. 11, 13 are a bit difficult to read – is it possible to increase font size?

Thanks for pointing this out, we have updated the figures with increased font size.

The abstract, introduction, and conclusion sections are quite repetitive, more or less listing the same results. Consider shortening the introduction and/or sharpening the abstract?

We thank the reviewer for this helpful suggestion. To address the repetition and avoid the redundancy across the above-mentioned sections, we have revised the introduction to provide more focused context and reduce the overlap with the abstract and the conclusion. We have also removed a redundant section in the (revised) discussion Sect. 4.

Line 1 and elsewhere: I would encourage different wording than "significant" when there is no statistical basis for the statements.

Good point: we have changed wording through the manuscript to avoid misunderstanding with "statistically significant".

Line 318: does this initialization also differ between regions or is it fixed?

The initialization of contrail ice mass and number is the same in all regions and it is only varied in study A. This should now be clearer with the newly added Table 4.

Line 335: it would be helpful with a bit more detail in this paper about what "typical background concentrations" mean in EMAC, for the different regions.

We thank the reviewer for this useful remark. To clarify, we have added a sentence in Sect. 2.3 to specify this: "*Background concentrations are provided as climatological means for all aerosol species simulated by MADE3, as well as for $SO_2$ and $OH$, as required by the online calculation of $H_2SO_4$*

*chemical production in Eq. (1). Typical values for these concentrations can be found in Figs. 7a, S13-S18 of Righi et al. (2023).*"

We have also included the initialisation data in the Zenodo record provided in the Data Availability section.

Line 342: If I understand correctly (otherwise, please clarify), this means that there is a range of estimates of the correction factors for each region? Given the sensitivity of the results to different parameters, this would seem like valuable information to add to the percentage numbers given throughout the text. Moreover, does "multiple ensemble simulations" here correspond to the number of ensembles given in table 1? If so, the number of ensembles differ substantially, does that affect the results?

Yes, your understanding is correct. The results in each region are given as ensemble averages and we agree with the suggestion about the variability within each ensemble. This information has been added to the text, by providing the +/- standard deviation whenever the percentage numbers for the plume correction are mentioned in the text.

Motivated by your question about the number of ensembles, we looked further into the data and analysed the distribution of all ensemble members for the REF(Wice) case. This is provided below for the number concentration in the SP and BG boxes, while the panel on the right shows the distribution at the reference time. The distribution is relatively narrow, especially for SP, with only a few outliers and a minimal difference between mean (blue) and median (red) at the reference time (box on the r.h.s. of each plot). Hence, we can expect the results to be quite stable against the number of ensemble members, although at least a few tens of ensembles are probably necessary to obtain a robust distribution.

[Figure]

The standard deviations of each plume correction further suggest that the properties of the background within a given ensemble rather than the ensemble size affect the variability of the results.

Line 366: "such as SO2" – what else? It does not seem that this study considers species other than sulfate and soot?

Correct, we have rephrased the sentence.

Line 406: what does "typical conditions" mean – annual means? Based on frequency or occurrence? Do the authors expect strong seasonal differences given the regional variation found? Please define/specify.

By "typical background conditions", we refer to the ambient meteorological conditions (temperature, pressure and relative humidity) and background aerosol concentrations at the cruise altitude for subsonic aircrafts. These values represent climatological means derived from EMAC simulations for different regions. We have clarified this in the text (see also our reply to a similar comment above).

Seasonal variations have not been considered, in order to generalize the applicability of the results for different regions and avoid a seasonal bias. Given that seasonal variations would affect temperature and humidity, we would expect this to have a potential impact on the frequency of the nucleation event, which could increase the variability of the ensemble means discussed in the above comment. This is an interesting aspect which should be explored in the context of global model applications of the plume model.

Line 449: Maybe a column could be added to table 2 given the initial background concentration in the different regions? (And temperature and RH differs in table 2 – these differences are included? Would these differences be expected to influence the results? Via sulfate production, but likely small effect…)

This would be difficult, as the background concentrations in the plume model need to be provided for all MADE3 tracers and modes, which results in more than 80 values (see also the reply about the comment to line 335 above). But we made this information available in the Zenodo record provided in the Data Availability section.

Line 551-554: It could maybe be good with some absolute numbers as well, not only percentage?

Thanks for this suggestion. The absolute numbers are small, as the plume model simulates the effect of a single aircraft, but we have included them anyway in the text wherever the plume corrections are mentioned.

Line 618: "remarkable" – is this the right word here?

We have rephrased this as follows: *"shows very small variations in the plume corrections"*

**Referee 2**

**No feedback to cloud microphysics or radiative forcing.** The paper discusses climate relevance, especially for low-level clouds, but does not actually quantify CCN changes or radiative impacts. Even a simplified treatment or discussion of how the aerosol differences would translate to cloud effects would enhance the paper's broader significance.

Following a similar comment by Referee #1, we have extended Sect. 4 (Discussion) to elaborate on the implications of the plume correction for aerosol number concentrations and the resulting climate effect. As discussed above in the reply to the first comment by Referee #1, to translate aerosol effects into cloud effects would require significant development beyond the current capabilities of the MADE3 scheme and is beyond the scope of this work. The plume model, however, has been designed having the (offline or online) coupling to global models in mind. This will allow to translate the aerosol effects into cloud effects and to quantify the effect of aviation-aerosol on low-level clouds taking the plume processes into account. For the reasons outlined in our response to Referee #1, we would refrain from speculating about the resulting climate effect in the present work, since a quantitative estimate is challenging without applying a global climate model.

**Scope of chemical processes.** The model only considers $H_2SO_4$ formation from $SO_2$ and neglects other chemical pathways or species (e.g., organics, $NO_x$, secondary organic aerosol formation). A clearer discussion of why this simplification is acceptable (or its implications) would help define the model's applicability.

Thank you for this suggestion, we have added the following paragraph at the end of Sect. 2.2.1 to discuss the implications of the simplified chemical process:

*"Previous studies have shown that the aviation effect on low clouds is largely driven by sulfate aerosol particles (Righi et al., 2013; Kapadia et al., 2016). Therefore, our study primarily focuses on sulfate aerosol, with a particular attention to sulfur chemistry. Although $NO_x$ is included in the plume model as a proxy for plume age, it is currently defined as a passive tracer and $NO_x$ chemistry is not considered. Given that it acts as a precursor for $HNO_3$ and aerosol nitrate but has no direct impact on particle number, this simplification is acceptable for the scope of this study. The role of organics, which in contrast could contribute to particle number via the nucleation of secondary organic aerosol (e.g., Liu and Matsui, 2022), is also not considered, due to the complexity of the involved chemistry, which would considerably increase the computational demand of the plume model. Nevertheless, as outlined in Sect. 4, the plume model is designed to allow coupling with global models and thus take advantage of the detailed chemical scheme of a host model to account for other chemical pathways."*

**Interfacing with a global model.** Since this model is designed for efficient use in global simulations, it would be helpful to briefly outline the strategy for integration into global models, and how the plume correction could be used as a parameterization.

We agree with this suggestion and we have extended Sect. 4 to include a brief description of a possible strategy for integrating the plume model into global models, as suggested. The inserted text reads as follows:

*"The plume model presented here is ready for application and can be coupled with global models, to calculate the changes in specific parameters of the aviation aerosol population such as mass, number and size, between the time of emission (i.e., the end of jet regime) and the end of the dispersion regime. This can provide the global models with refined and more physically motivated assumptions to initialise aviation emissions in global simulations, considering the plume-scale processes that these models cannot resolve. The aviation emission inventories required to initialise global aerosol-climate models for assessing the climate impact of aviation usually provide the mass of emitted species, but do not include particle number emissions, especially for volatile particles that are relevant for the aviation impact on low clouds. Converting mass to particle number requires assumptions about the particle size distribution at emission. Current methods are hampered by the scarcity of observational data at different plume stages, especially on particle number and size, which introduces uncertainty and potential biases in representing aviation-induced aerosols. This challenge is compounded by the fact that particle size distributions evolve over time, with younger plumes typically containing larger concentrations of smaller-sized particles, but global models cannot resolve this evolution due to their coarse resolution. The plume model presented here could also suffer from the scarcity of available observational data for initialisation and shows high sensitivity to some of these initial parameters, but it effectively helps to bridge the gap between the beginning of the vortex regime and the large scale of global models, producing detailed output in terms of species mass, and number concentrations categorized by size mode and mixing states at different stages of the plume evolution. From these results, number-to-mass ratios can be derived for each aerosol mode and used to calculate more physically based emitted particle numbers. Given the low computational costs of the plume model, it can also be employed as an online parametrisation, thereby more accurately accounting for the impact of local conditions and detailed background chemistry."*

**Limitations of modal models regarding size distribution predictions.** While the authors note that the MADE3 microphysics scheme has been evaluated and found to perform well in global-scale applications, I would encourage them to consider the limitations of applying modal aerosol models in plume-resolved simulations. The physical and chemical conditions in aircraft exhaust plumes—such as steep concentration gradients, rapid dilution, and intense nucleation—differ markedly from the large-scale, averaged environments for which these schemes were originally designed. Recent work by Fierce et al. (J. Aerosol Sci., 181, 106388, 2024) highlights significant discrepancies between the modal MAM4 scheme and particle-resolved size distributions from PartMC, particularly under plume-like conditions. This raises important questions about the suitability of modal representations for capturing the microphysical evolution of aerosol populations in near-field plume simulations. I recommend that the authors discuss these limitations more explicitly and consider how they might affect the accuracy of their results and conclusions.

We acknowledge the limitations of applying a modal scheme for plume-resolved simulations and have added a point to the list of model limitations in Sect. 4 to mention this issue. We note, however, that although absolute difference between a modal scheme and a particle resolving model will certainly arise, what matters here is the difference between the plume approach and the instantaneous dispersion approach, which has been demonstrated in this work together with its regional and parametric sensitivities. Another reason for choosing a modal scheme was to allow for the possibility to implement our plume model directly into global models, by means of an online parametrization. Given the computational demands, this would not be feasible with a more detailed aerosol scheme or a particle resolving model. Using a particle resolving model to develop an (offline)

plume parametrization could indeed be an interesting approach and could be considered for future studies, also based on the knowledge gained in the current work.

**Presentation of the governing equations.** The presentation of the governing equations would benefit from greater clarity and completeness. Currently, mathematical expressions are scattered throughout the text, but they mostly appear to be discretized or implementation-specific snippets rather than a concise formulation of the core model equations. To aid reader understanding and improve transparency, I recommend that the authors clearly define the governing equations at the outset of the methods section, beginning with a clear statement of the state variables (e.g., aerosol number, mass, moments, gas-phase species) and their dependencies (e.g., time, space, size). A brief but coherent listing of the continuous (pre-discretization) equations governing aerosol dynamics, gas-phase chemistry, and aerosol–gas interactions would greatly improve the readability and reproducibility of the study. While this level of detail is sometimes omitted in our field, including it here would strengthen the paper's clarity and accessibility, particularly for a paper published in GMD.

As our study focuses primarily on the plume model processes, the corresponding governing equations are already present in the paper (Sect. 2.3). The equations governing the aerosol dynamics, gas-phase chemistry, and aerosol-gas interactions are indeed fundamental to our work, however these equations have been comprehensively documented in the earlier publications (Kaiser et al., 2014; Kaiser et al., 2019), and have not been modified for this study. As the paper is already quite long and detailed, we think that including additional equations would reduce the readability. We also note that the code is available via Zenodo in the Code Availability section. This, together with the information provided in the manuscript and the initialisation data given in the Data Availability section, ensures full reproducibility of the results.

**Comparison to measurements.** While I recognize that this is a process-level plume model and not intended to reproduce specific observations, I think it is still important to address the question of model validation or evaluation. Ultimately, we would want to know whether applying this modeling framework leads to improved predictions—either of aerosol properties in the far field, or of emergent metrics like CCN or optical properties. I understand that direct comparison with measurements may not be feasible at this stage, but some discussion of how the model's outputs could eventually be evaluated—e.g., through comparison with aircraft observations, or as part of a nested modeling framework—would strengthen the manuscript. Even a statement clarifying that this work is a foundational step toward such evaluation would help position the contribution more clearly.

Thank you for raising this important point. We fully agree that model evaluation is a critical step toward establishing the predictive capability of any plume modelling framework. While the current study focuses on the process-level details and description of the plume model, with the goal of implementing it as a parametrization in global models, we have added the following paragraph to Sect. 4 (Discussion) with the suggestion for possible validation:

*"Given these limitations, a direct evaluation of the plume model result against observational data is challenging, also due to the fact that direct measurements in the very early plume stages are difficult to achieve due to safety constraints and instrumentation limitations, especially in the presence of contrail ice. The model output for the aged plume remains suitable for evaluation and can be compared against in-situ data from aircraft campaigns. While such measurements are inherently*

*limited in spatial coverage and plume positioning, $CO_2$ and $NO_x$ concentrations can be used as proxies for plume age, and our model is able to provide consistent plume age estimates based on these tracers, as shown in Mahnke et al. (2024). A further option would be to compare against large-scale in situ measurements as provided, for instance, by IAGOS (Brenninkmeijer et al., 2007): this, however, would require extending the model to account for multiple plumes, and for their interaction and superposition across various geometries. Besides evaluation purposes, future in situ measurements targeting aircraft plumes could also provide valuable data for model initialisation, especially concerning the size-resolved properties of volatile nucleation mode particles (Dischl et al., 2024, Williamson et al., 2018)."*

Equation 1: l.h.s (rate) and r.h.s. (concentration) are inconsistent.

We have corrected the equation, thanks for spotting.

L. 226: "only" appears twice.

Text corrected.

L. 232: "gets" is very colloquial. Suggest "is".

Agree, we have changed it as suggested (there were two occurrences in the paper).

L. 250: "same size" -> same mode

Corrected.

L. 276: value of s is unclear – should this be 0.004 s⁻¹?

Thanks for noting this, the units have been added.

Equation 8: C_i^SP(t) appears on the r.h.s. – is this correct?

This is indeed confusing: we used the same symbol as in the code this variable is overwritten with the same name, but mathematically this is of course not correct. We have changed the symbol and now indicate the updated concentration after entrainment with $\hat{C}_i^{SP}(t)$.

L. 311: should be ice crystal number *concentration*. This happens quite often throughout the paper, please check throughout.

Checked and changed in the whole document.

L. 418: do you mean: "in terms of total mass and number concentration, and number size distribution"

Yes, we have added this for more clarity. Note, however, that this is not "total", since for mass we look at individual species (sulfate and soot) and different modes.

Figure axes labels are too small in Figure 11 and 13.

Thanks for highlighting this: we have increased the font size, following the same remark by Reviewer#1.